# Minor Forms of Parental Maltreatment and Educational Achievement of Immigrant Youths in Young Adulthood: A Longitudinal Study

**DOI:** 10.3390/ijerph20010873

**Published:** 2023-01-03

**Authors:** Jerf W. K. Yeung, Hui-Fang Chen, Herman H. M. Lo, Leilei Xu, Chi Xu

**Affiliations:** 1Department of Social and Behavioural Sciences, City University of Hong Kong, Kowloon, Hong Kong, China; 2Professional Practice and Assessment Centre, Department of Applied Social Sciences, The Hong Kong Polytechnic University, Kowloon, Hong Kong, China

**Keywords:** parental hostility, emotional rejection, future academic aspirations, immigrant youths, college graduation

## Abstract

Parental hostility and emotional rejection—or aggregated as general harsh family interactions with parents—have received little research attention due to such parent-child interactions being counted as minor forms of parental maltreatment and regarded as being less harmful. However, recent research showed that these minor forms of parental maltreatment on youth development are far from negligibility on account of their frequency, chronicity, and incessancy. In this longitudinal study, we investigated how parental hostility, emotional rejection, and harsh family interactions with parents of in early adolescence of immigrant youths (wave-1 M_age_ = 14) adversely impact successful college graduation of immigrant youths in young adulthood (wave-3 M_age_ = 24) through the mediation of their development of academic aspirations in late adolescence (wave-2 M_age_ = 17). Using data from a representative sample of the Children of Immigrants Longitudinal Study (*N* = 3344), the current study revealed that parental hostility, emotional rejection, and harsh family interactions with parents significantly impaired successful college graduation of immigrant youths in young adulthood, with the decreased odds of 20.1% to 30.22%. Furthermore, academic aspirations of immigrant youths in late adolescence not only significantly mediated the abovementioned relationships but also contributed to the higher odds of immigrant youths’ college graduation by 2.226 to 2.257 times. Findings of this study related to educational innovations, family services, and policy implications are discussed herein.

## 1. Introduction

Adolescence is a critical formative and transitional period for youths to prepare for adulthood [1,2], in which family interactions and socialization experiences with parents are profoundly influential on their various aspects of development. This is valid, as family has been reported to be the most intimate and proximal socialization agent to shape youth development [3,4]. Abundant research evidence supports the adverse impacts of harsh parenting and parental maltreatment on behavioral problems, psychological difficulties, maladaptation, delinquency, and educational problems of youths [5,6,7]. Parental hostility, manifesting as conflicting and clashing parental attitudes and reactions toward their children, as well as parental emotional rejection, indicating parental displays of dislike and disdain toward their children, are common minor forms of parental maltreatment and harsh discipline [6,8]. In fact, parental hostility and emotional rejection are generally indicative harsh family interactions with parents in the family realm experienced by youths [5,6]. Nevertheless, research on how harsh family interactions with parents negatively affect long-term development of youths is lacking, especially for immigrant youths regarding the importance of their educational achievement in adulthood [9,10].

Immigrant youths, compared to their non-immigrant counterparts, are at greater risk of being harshly disciplined and maltreated by their parents due to lower parental education levels, disadvantaged social status, and cultural differences [11]. This is correspondent with what Romano et al. [12] mentioned, i.e., that “socioeconomic disadvantage is important because such stressful conditions as poverty often compromise effective parenting practices (p. 433)”. However, no longitudinal research has been conducted to investigate the negative effects of parental hostility, emotional rejection, and their aggregate as general harsh family interactions with parents experienced by immigrant youths on their educational success in young adulthood. This is important, as obtaining a college degree provides immigrant youths with a chance of upward social mobility; for example, Portes et al. [13] mentioned “(w)ithout the costly and time-consuming achievement of a university degree, such dreams are likely to remain beyond reach (p. 1081)”. Furthermore, some research evidence shows that harsh parenting and parental maltreatment first impair youths’ self-concept and hopeful future, in turn hampering their later academic, behavioral, and psychological development [14,15]. Accordingly, in this study, we attempted to investigate how parental hostility, emotional rejection, and harsh family interactions with parents in early adolescence may adversely compromise successful college graduation of immigrant youths in young adulthood through the mediation of their development of academic aspirations in late adolescence [16,17].

### 1.1. Harsh Family Interactions with Parents and Educational Achievement

Harsh parenting and parental maltreatment can occur in various forms and types, which are commonly referred to parents or main caregivers using abusive, hostile, disdainful, and neglectful responses toward their offspring behaviorally, psychologically, and emotionally [6,7,18]. Empirical research has shown that harshly parented and maltreated youths experience manifest cognitive, psychological, and behavioral maladjustment [5,8] and perform poorer academically and have greater school difficulties [12,19,20]. This is valid, as harsh parenting and parental maltreatment represent a maladaptive style of family socialization and parent–child relationships characterizing low parental love, approval, acceptance, and support [21,22], which may profoundly impair the attachment and security needs of children and youths for positive development, thereby leading to their poorer social and educational adjustment [6,7]. Nevertheless, parental hostility and emotional rejection—or aggregated as a general form of harsh family interactions with parents in the family—have received inadequate research attention, mainly due to their being regarded as a minor form of parental maltreatment or just harsh parental discipline when compared with physical and sexual abuse [23,24]. This corresponds to what Allan et al. [25] mentioned, i.e., that “parental hostility, as opposed to physical or sexual abuse and neglect, has not received much attention. …Abuse generally consists of physical or sexual brutality whereas hostility generally consists of verbal and psychological abusiveness such as anger or antagonism (p. 169)”. In addition, Khaleque [26] stated that “(p)arental (emotional) rejection, on the other hand, refers to the withdrawal or lack of parental warmth, affection, care, comfort, nurturance, support, or love toward their children (p. 977).” Hence, it is justified to conceptualize parental hostility in this study as the manifestation of conflicting and clashing relationships between parents and their young children [25,27]; parental emotional rejection as parental displays of a lack of interest, warmth, love, and nurturance toward their young children in an aversive and disapproving way [28,29]; and harsh family interactions with parents as the aggregate of these conflicting, clashing, rejecting, apathetic, and disapproving processes of parent–youth interactions [25,28,30].

Pertinently, the family transmission model evinces that parents are the main socialization figures responsible for providing care, support, resources, and the social and cultural capital necessary for the healthy cognitive development of their offspring [3]. However, harshly disciplined and maltreated youths who have experienced minor forms of parental hostility, emotional rejection, or harsh family interactions with parents in the family imply their receiving inadequate parental educational involvement and support and necessary learning resources and guidance to help them overcome academic challenges for educational success [11,12]. This is especially important for immigrant youths, as they generally live under the conditions of family poverty and weak social supports and limited resources [31,32]. Although some related research has reported the harmful impacts of parental hostility and rejection on behavioral and psychological adjustment of children and youths, little is known about how these negative family socialization experiences may negatively affect the educational development of immigrant youths. Resonantly, the model of family socialization indicates that youths experiencing harsh discipline and maltreatment undergo low levels of parental responsiveness and support and high levels of parental negativity [22,33,34], which are harmful to their constructive educational and social development. In a meta-analysis, Khaleque [26] found that paternal hostility/aggression is significantly associated with psychological maladjustment and negative personality dispositions of children across ethnicities, cultures, and geographical boundaries. Similarly, in their latest study, Backman, Laajasalo, Jokela, and Aronen [23] found that parental hostility and low levels of maternal warmth were significantly predictive of psychopathic behaviors of offending adolescents. More recently, Lee and Mun [35] reported that parental rejection was positively related to cyberbullying behaviors of Korean children and youths in a representative sample from the 2019 Korean Children and Youth Panel Survey (KCYPS); this relationship was independent and sequentially mediated by children’s depression and smartphone addiction. Putnick et al. [8] found that parental rejection was longitudinally predictive of lower school scores among primary school children across nine countries. Moreover, Ryan et al. [20] investigated 732,828 youths born between 2000 and 2006 in Michigan public schools and found that those experiencing child protective service (CPS) involvement exhibited significantly lower math and reading scores, grade retention, and receipt of special education than their peers without CPS involvement. Furthermore, Welsh et al. [36] found that college students who experienced a history of maltreatment exhibited significantly lower first-semester GPAs and poorer college adaptation.

Accordingly, as immigrant parents are of disadvantaged social status, economic deprivation, and cultural differences, they may be more prone to employ harsh discipline and unsupportive parenting [11,27,37]. Therefore, it is expected that experiences of parental hostility, emotional rejection, and/or harsh family interactions with parents in the family are more common for immigrant youths, which is believed to detriment their educational achievement in adulthood, for example, successful college graduation with a four-year undergraduate degree. Due to the more disadvantaged and difficult family socialization environment of immigrant youths and the importance of educational success for their social mobility in adulthood as compared to their better-off local counterparts [38,39], it is theoretically and empirically important to study how immigrant youths’ experiences of parental hostility, emotional rejection, and harsh family interactions with parents in early adolescence may adversely affect their later successful college graduation in young adulthood. In addition, immigrant youths’ experiences of parental hostility and emotional rejection may occur independently or jointly in the form of harsh family interactions with parents in the family, which is congruous with the claim that concurrent exposure to multiple types of harsh parental discipline and maltreatment could further impair youth development [36]. Hence, it is justifiable to examine the individual and combined effects of parental hostility, emotional rejection, and harsh family interactions with parents on the educational achievement of immigrant youths in adulthood.

### 1.2. The Mediation of Academic Aspirations of Immigrant Youths

During adolescence, youths may ask the questions, “Who am I?” and “What will I do in the future?”, which are closely related to their formation of self-concept and identity, a process referring to the establishment of the “possible self” in relation to the future [40]. Youths who have been harshly disciplined and maltreated may develop a sense of worthlessness, inferiority, and incapability [7], which may directly compromise their future hope and expectations for academic and social success [7,20]. Consistently, the self-system theory of motivational development posits that the general self of youths is socially constructed, especially through the process of family socialization and experiences, in which youths’ socially constructed self may act as the cognitive and motivational foundation for them to pursue future goals and success [17,41]. If youths have been harshly parented and maltreated, including experiencing parental hostility, emotional rejection, and harsh family interactions with parents, they tend to hold a negative view of themselves, which may severely undermine their aspirations for future educational and social success [5,7].

As academic aspirations of youths are mainly shaped by the process of family socialization and experiences in early years [14,15], it hence is believed that the academic aspirations of immigrant youths in late adolescence may not only directly contribute to their educational achievement in young adulthood [12,41,42] but also mediate the effects of parental hostility, emotional rejection, and harsh family interactions with parents on their later educational success in young adulthood [17,38]. In a longitudinal study, Hentges and Wang [14] found that harsh parenting in 7th grade was significantly related to lower GPAs among students in 11th grade through the mediation of compromised academic values in 8th grade. Moreover, Seginer and Mahajna [17] reported that perceived positive parenting significantly and positively predicated Muslim youths’ academic aspirations for higher education in 11th grade, which, in turn, mediated the relationship between parenting and youths’ academic performance. In her qualitative study, Morton [15] explored how maltreatment experiences of foster youths might result in cognitive and behavioral difficulties in relation to their compromised educational achievement and found that maltreated youths generally held lower academic aspirations, seriously harming their success in college education. Therefore, in this study, we expected that immigrant youths’ experiences of parental hostility, emotional rejection, and harsh family interactions with parents in early adolescence would negatively impair their development of academic aspirations in late adolescence and successful college graduation in young adulthood, in which the development of academic aspirations in late adolescence would mediate the effects of parental hostility, emotional rejection, and harsh family interactions with parents in early adolescence of immigrant youths on their successful college graduation in young adulthood.

### 1.3. Contextual Influences and Educational Achievement of Immigrant Youths

Contextual differences of the living environment can undeniably have distinct structural effects on youth development, including educational success. However, prior research has seldom considered contextual influences on educational achievement of both immigrant and non-immigrant youths when studying harsh parental discipline and maltreatment, such as parental hostility, emotional rejection, and harsh family interactions with parents. This is important because youths’ educational achievement is highly susceptible to their living contexts of their family, cultural, and school environments [43,44], which is consonant with the social systems theory proposing that individual developmental outcomes are simultaneously shaped by multiple social systems at different levels [45]. As youth development deeply hinges on family socialization, cultural influences, and school cultivation, in this study, we attempted to include the contextual influences of family, cultural, and school environments when examining the effects of parental hostility, emotional rejection, and harsh family interactions with parents on successful college graduation of immigrant youths in young adulthood. Based on the segmented assimilation theory, immigration scholars have proposed that family composition, parental socioeconomic status (SES), cultural mode of reception, school type, school minority status, and school location are all critically influential on educational development of immigrant youths [13,42]. Immigration research has shown that immigrant youths from two-parent families, of higher parental SES, and not receiving negative reception perform better both educationally and socially [32,42,46]. In addition, empirical evidence shows that immigrant youths studying in public, minority-receiving, and inner-city schools had poorer educational development than their counterparts in private, native, and suburban schools [13,46]. Hence, in this study, we incorporated the contextual effects of family composition, parental SES, and mode of reception at the individual level, and school type, school minority status, and school location at the school level as covariates when examining the relationships of parental hostility, emotional rejection, and harsh family interactions with parents in contribution to academic aspirations of immigrant youths in late adolescence and their successful college graduation in young adulthood.

## 2. The Present Study

The aim of this study was to investigate the effects of immigrant youths’ experiences of parental hostility, emotional rejection, and harsh family interactions with parents—aggregated as parental hostility and emotional rejection—in early adolescence on their successful college graduation in young adulthood through the mediation of immigrant youths’ academic aspirations in late adolescence. This study builds on prior pertinent research with the following advantages. First, it is a longitudinal study covering a long-range time period across the 10-year life course development of immigrant youths from early adolescence to young adulthood. This is valuable, as prior studies mainly limited their focus on educational performance of children and youths in primary or high school years [7,12,14]. Secondly, examining the mediation of immigrant youths’ academic aspirations has profound social and policy implications because academic aspirations of youths are a malleable and cultivable intrapersonal cognitive trait [40,41] that can aid in the design and delivery of beneficial educational services and family interventions to assist immigrant youths and their families in need. Thirdly, elucidating the process of how early harsh family socialization experiences of immigrant youths impact their educational success in young adulthood may benefit the immigrant population as a whole because obtaining a college degree signals a milestone for immigrant youths to change their life trajectories, indicating the possibility of upward social mobility [13,46]. Fourthly, harsh discipline and strict parenting are more common in immigrant families than non-immigrant families [9,32], which deserves more research to examine the effects of these undesirable family experiences on educational development of immigrant youths in adulthood. Taken together, we posited the following hypotheses:

**H1.** 
*Parental hostility, emotional rejection, and harsh family interactions with parents—aggregated as parental hostility and emotional rejection—in early adolescence of immigrant youths would negatively predict their successful college graduation in young adulthood.*


**H2.** 
*Parental hostility, emotional rejection, and harsh family interactions with parents in early adolescence of immigrant youths would negatively predict their academic aspirations in late adolescence, even controlling their initial level of academic aspirations in early adolescence.*


**H3.** 
*Academic aspirations of immigrant youths in late adolescence would mediate the effects of parental hostility, emotional rejection, and harsh family interactions with parents in early adolescence of immigrant youths on their successful college graduation in young adulthood.*


**H4.** 
*Compared to parental hostility and emotional rejection, harsh family interactions with parents in early adolescence of immigrant youths would more strongly predict their academic aspirations in late adolescence and successful college graduation in young adulthood.*


In order to preclude confounding effects, in this study, we adjusted for gender, age, number of siblings, generation status, standardized English and math test scores, and ethnic origins of immigrant youths. Immigration research indicates that female immigrant youths generally perform better academically than their male counterparts [47] and that older immigrant youths have more difficulties in adjusting to school than their younger counterparts [32]. Furthermore, immigrant youths with more siblings are likely to experience diluted family supports and care, which may compromise their school performance [32,42]. With respect to immigrant generation status, earlier generations of immigrant youths are reported to have higher educational motivation and performance than later generations, despite their economic and social disadvantages, which is known as the “generation mystery” [48]. Additionally, the ethnic origins of immigrant youths have been found to affect their educational achievement; immigrant youths of Northeast/Eastern Asian origin showed higher educational success, and Mexican and Caribbean immigrant youths manifestly fell behind academically, whereas other ethnic groups fell in between these two groups in terms of educational achievement [42,49]. Therefore, in this study, immigrant youths are classified into eight major ethnic groups with reference to their cultural and geographic adjacency [32,49]: Mexican, Caribbean, Cuban, Central and South American, Southeast Asian, Northeast/Eastern Asian, Middle Eastern and African, and European origins.

## 3. Materials and Methods

### 3.1. Sample and Procedures

The data used in this longitudinal study were obtained from the Children of Immigrants Longitudinal Study (CILS), which is one of the largest longitudinal surveys on the development and adjustment of immigrant youths conducted in the United States [50]. CILS recruited participants of immigrant youths from two major immigrant-receiving regions of the country, Miami/Ft. Lauderdale and San Diego. The sample of immigrant youths included both U.S.-born and foreign-born children with at least one immigrant parent. The wave-1 survey of CILS was conducted in 1992 with a school-based sampling frame, which involved interviews of 5262 immigrant youth participants ($\bar{X}$_age_ = 14 years old) in 8th and 9th grades studying in 49 public and private schools, representative of the population of immigrant youths in the study regions. In 1995–1996, the wave-2 survey of CILS involved reinterviewing 4288 immigrant youths (81.5%) of the original sample at the time of their high school graduation ($\bar{X}$_age_ = 17 years old). In 2002, the wave-3 survey of CILS was conducted when the immigrant youths were in their young adulthood ($\bar{X}$_age_ = 24 years old), including a sample of 3613 immigrant youths representing 68.9% and 84.3% of the wave-1 and 2 samples, respectively. In the present study, we used data across the three waves of CILS covering a 10-year life course of immigrant youths’ adaptation and development. Data analysis in the present study was based on a sample of 3344 immigrant youths who had given information regarding the study variables concerned. The rates of missing values were very low in the drawn sample of 3344 immigrant youths, ranging from 0.6% to 0.9%. The missing values were largely due to missing responses of immigrant youths to one of the question items used to construct the composite scores of parental emotional rejection and academic aspirations. The technique of expectation maximization imputation (EM) was used to replace the missing values rather than employing mean substitution, as the former involves a two-step iterative process whereby regression analysis and maximum likelihood are used to impute missing values through simulation with reference to all the available data as predictions to effectively avoid the problems of restricted variances and biased estimates associated with mean substitution [51].

### 3.2. Measures

Successful college graduation of immigrant youths in young adulthood was measured with a single question item asking whether the CILS immigrant youth participants had graduated from a college with a four-year undergraduate degree at the time of the wave-3 CILS survey. A dichotomous response to the question item was used, in which 1 = yes and 0 = no.

Academic aspirations of immigrant youths were measured by two question items included in the wave-1 and wave-2 CILS surveys, which asked the immigrant youth (1) the highest education level he/she would like to achieve and (2) the highest education level he/she thinks they would realistically achieve. A 5-point scale was used to rate participants’ responses to the two items, ranging from 1 = less than high school to 5 = finish a graduate degree. The two items were averaged in the wave-1 and wave-2 CILS surveys to represent the academic aspirations of immigrant youths in early and late adolescence, respectively [16,42]; higher scores indicate higher future academic expectations and motivations. Di Giunta et al. [16] used similar question items to measure youths’ academic aspirations, proving the external validity of these questions. Cronbach alpha coefficients of academic aspirations of immigrant youths in the wave-1 and wave-2 CILS surveys were adequate: α = 0.805 and 0.815, respectively.

Parental hostility of immigrant youths in early adolescence was measured in the wave-1 CILS survey, in which immigrant youths were asked whether the processes of family socialization and communication with parents were full of clashes and hostility from their parents. The response was provided on a 4-point scale, where 1 = all of the time, 2 = most of the time, 3 = sometimes, and 4 = never, which was inversely coded, so higher scores represent more parental hostility. Although it is more desirable to employ validated scales to measure parental hostility, such as the Parental Acceptance–Rejection Questionnaire [52], CILS is a large longitudinal survey of the life course of immigrant youths that makes it impossible to incorporate multi-item scales to measure single behaviors or perceptions in trade-off of parsimony and avoidance of attrition [53]; this is common in large-scale longitudinal surveys. Nevertheless, recent empirical research has adopted a similar approach to measure parental hostility and aggression in relation to adolescent delinquency and substance use [27], which appears to be methodologically valid and reliable.

Parental emotional rejection of immigrant youths in early adolescence was measured with two items in the wave-1 CILS survey asking immigrant youths (1) whether their parents did not like him/her much and (2) whether their parents were not interested in him/her; these items were rated on a 4-point scale, where 1 = very true, 2 = partly true, 3 = not very true, and 4 = not true at all. Again, although the use of validated scales to measure parental emotional rejection is more informative, the latest relevant studies used similar items comparable to the items adopted in the current study to measure parental emotional rejection [18,37], supporting its reliability and validity. The correlation coefficient of the two items was *r* = 0.439, *p* < 0.001, and the Cronbach alpha was α = 0.611, representing an adequate level.

Harsh family interactions with parents in early adolescence of immigrant youths were measured by combining the items used to indicate the aggregate of parental hostility and emotional rejection. This is justified, as the indicators and dimensions of parental harsh discipline and maltreatment were found to be closely interrelated, with more harmful effects on child and youth development when they occurred in combination [6,26,37]. Because harsh family interactions with parents were measured jointly by loading the indicators of parental hostility and emotional rejection together, composite reliability was used to report its internal consistency, which was adequate, *pc* = 0.600. In addition, the average correlation coefficient among the three items was adequate: *r* = 0.316, *p* < 0.001. Hence, harsh family interactions with parents were measured by combining the three items as a composite score.

For contextual variables obtained from the wave-1 CILS survey, family composition was measured as a dummy variable with reference to whether the immigrant youths were living with both their biological father and mother (1 = two-parent family and 0 = otherwise). Parental SES was included as a unit-weighted standardized variable by averaging occupational socioeconomic index, parental education, and home ownership, with higher scores representing better socioeconomic levels [32]. Mode of reception indicates negative or position/neutral incorporation that the immigrant youth received due to his/her ethnic origin (1 = negative, 0 = positive/neutral); immigrant youths of Haitian, Jamaican/West Indian, Mexican, or Nicaraguan origin were assigned to negative reception [13]. Moreover, school type was included as a dichotomous variable, referring to immigrant youths attending public or private schools (1 = public school, 0 = private school); school minority status was counted if 60% or more of the students in a school were of ethnic minority status (1 = minority school, 0 = otherwise), and school location indicated whether the school was situated in an inner-city or suburban area (1 = inner-city, 0 = otherwise).

For sociodemographic variables of immigrant youths, gender was measured as a dummy variable (female = 1, 0 = male), and age and number of siblings were included as count variables. In addition, immigrant generation status was classified as 1.5 generation (immigrated with both parents to the U.S.), 2 generation (born in the U.S. with two immigrant parents), or 2.5 generation (born in the U.S. with one immigrant parent and one native parent) served as a reference [47]. Standardized English and math test scores were based on the scores of standardized the Stanford Achievement Test provided by the schools participating in wave-1 CILS. Ethnicities of immigrant youths included Mexican, Caribbean, Cuban, Central and South American, Southeast Asian, Middle Eastern and African, and European origins; Northeast/Eastern Asian immigrant youths served as comparison group due to their academic outperformance [46,49].

### 3.3. Analytical Procedures

Due to the clustered nature of the CILS data nested in schools, multilevel modeling was applied; a generalized linear mixed modeling (GLMM) approach was employed because successful college graduation of immigrant youths in young adulthood was a dichotomous outcome, which can express as:g(*μ_i_*) = *η_i_*,(1)
where the link function of g(.) is to convert the expected value (*μ_i_*) to *Y_i_* in a function of the linear predictor (*η_i_*). Therefore, we can have:*μ_i_* = E(*Y_i_*).(2)

For this, GLMM generates the form as
(3)$$h^{-1}\left\{E(y_{ij}|\varsigma_{j},x_{ij},z_{ij})\right\}\;=\;x_{ij}^{\prime }\beta +z_{ij}^{\prime }\varsigma_{j}\equiv \eta_{ij}$$
where $h^{-1}\left(.\right)$ is the link function of $x_{ij}^{\prime }\beta +z_{ij}^{\prime }\varsigma_{j}$, and $\eta_{ij}\;$is the linear predictor meaning a concrete definition. As such, the response is assumed to be independent and has conditional distributions from the exponential family provided with the covariates and random effects. Accordingly, we can present the following equation of generalized linear mixed modeling to predict successful college graduation of immigrant youths in young adulthood, which is:$$\begin{array}{cc} \eta_{ij}= & \gamma_{00}+\gamma_{01}PS_{j}+\gamma_{02}MS_{j}+\gamma_{03}IS_{j}\\ & +\;\beta_{1}PH_{ij}+\beta_{2}PR_{ij}+\beta_{3}HF_{ij}+\beta_{4}AA_{ij}\\ & +X\beta CS_{ij}+\zeta_{0j}+\varepsilon_{i}, \end{array}$$
where $\eta_{ij}$ is the outcome variable of immigrant youths’ successful college graduation in young adulthood; $\gamma_{00}$ is the fixed intercept at the school level; and $\gamma_{01}PS_{j}$, $\gamma_{02}MS_{j},\;\mathrm{and}\;\gamma_{03}IS_{j}$ are school-level predictors of public school, minority school, and inner-city school status, respectively. In addition, $\beta_{1}PH_{ij}$, $\beta_{2}PR_{ij}$, and $\beta_{3}HF_{ij}$ refer to parental hostility, parental emotional rejection, and harsh family interactions with parents, respectively, at the individual level; and $\beta_{4}AA_{ij}$ is immigrant youths’ academic aspirations in late adolescence, which also acts as a mediator in later modeling procedures. Moreover, $X\beta CS_{ij}\;$represents a matrix for the sociodemographic covariates of immigrant youths’ gender, age, number of siblings, generation status, standardized English and math test scores, and ethnic origins in an abbreviated short form; and $\zeta_{0j}$ and $\varepsilon_{i}\;$are a random intercept and residual, respectively, at the individual level. In order to test the mediated effects of immigrant youths’ academic aspirations in late adolescence, model constraints were employed by setting new indirect paths, which were required because common mediational tests cannot be feasibly performed in multilevel modeling procedures [54]. The modeling procedures were conducted using *M*plus 8.5 [55].

## 4. Results

Table 1 shows the demographic and background statistics of the immigrant youths (*N* = 3344) who took part in the three waves of the CILS. Slightly more female immigrant youths participated than their male counterparts, accounting for 54.1% of the total sample (*n* = 1809). The mean age of immigrant youths in the wave-1 survey was 14.23 years, with an average of 1.79 siblings in the home. Moreover, 1.5-generation (46.9%, *n* = 1567) and 2-generation (41.3%, *n* = 1380) immigrant youths were more represented in the sample than their 2.5-generation counterparts (11.9%, *n* = 397). The standardized English and math test scores of the participants averaged 700.762 and 669.111, respectively. In terms of family composition, approximately one-quarter of the immigrant youths came from non-two-parent households (24.2%, *n* = 810), and the average score of parental SES, calculated according to the Duncan Socioeconomic Index (SEI), was 34.25 that is much lower than the means of the American public obtained from the General Social Survey of 1994 [56], which ranged between 44.2 and 48.7. Moreover, more than one-quarter of immigrant youths received negative reception due to their ethnicity (27.1%, *n* = 906). In terms of ethnic origins, Cuban and Southeast Asian immigrant youths were most prevalent in the sample (24.9% and 28.9%, respectively), and immigrant youths from Northeast/East Asia (2.3%), the Middle East/Africa (1.9%), and Europe (1.9%) were the least represented in the sample, with immigrant youths of Central/South American (16.6%), Mexican (12.4%), and Caribbean (10.9%) origins falling in the middle.

As the current study was based on the drawn sample of 3344 immigrant youths who provided relevant information regarding the study variables across the three waves of the CILS, it is important to compare their differences with the original sample of 5262 immigrant youth participants in terms of sociodemographic and contextual variables obtained in the wave-1 CILS survey, although we cannot examine their differences in terms of the main study variables due to attrition of the original sample. Independent samples bootstrapping *t*-tests conducted via the biased-corrected and accelerated (*BCa*) method (*ĵ* = 1000) showed that the drawn and original samples did not differ in terms of age (*t* = 1.978, *p* = 0.058, 95%*CIs* = −0.010 to 0.112), number of siblings (*t* = 0.691, *p* = 0.490, 95%*CIs* = −0.047 to 0.975), immigrant generations (*t* = −0.717, *p* = 0.474, 95%*CIs* = −0.060 to 0.030), standardized English scores (*t* = −1.476, *p* = 0.130, 95%*CIs* = −6.705 to 0.974), standardized math scores (*t* = −1.934, *p* = 0.520, 95%*CIs* = −7.672 to 0.041), or parental SES (*t* = −1.250, *p* = 0.202, 95%*CIs* = −1.187 to 0.396). In addition, Chi-square bootstrapping tests conducted via the biased-corrected and accelerated (*BCa*) method (*ĵ* = 1000) revealed that compared to the original sample, the drawn sample comprised slightly higher proportions of female immigrant youths (54.1% vs. 51.1%; *X*^2^(1) = 6.805, *p* < 0.01, 95%*CIs* = 0.007 to 0.051), immigrant youths from two-parent families (70.0% vs. 64.7%; *X*^2^(1) = 23.577, *p* < 0.001, 95%*CIs* = 0.033 to 0.075), positive/neutral reception (92.4% vs. 90.5%; *X*^2^(1) = 8.426, *p* < 0.01, 95%*CIs* = 0.011 to 0.055), and immigrant youths attending suburban schools (67.2% vs. 63.1%; *X*^2^(1) = 13.719 (1), *p* < 0.001, 95%*CIs* = 0.019 to 0.064). Nevertheless, Cohen’s effects sizes showed that the significant differences were minimal, ranging from *ω* = 0.029 to 0.054. Moreover, Chi-square bootstrapping tests conducted via the biased-corrected and accelerated (*BCa*) method (*ĵ* = 1000) did not reveal significant differences between the drawn and original samples in terms of ethnic origins (*X*^2^(7) = 9.080, *p* = 0.247, 95%*CIs* = 0.000 to 0.034), school type (*X*^2^(1) = 1.910, *p* = 0.167, 95%*CIs* = 0.000 to 0.037), or minority school status (*X*^2^(1) = 0.367, *p* = 0.545, 95%*CIs* = 0.000 to 0.028). Accordingly, there is no solid evidence to show the differences between the drawn and original samples. Nevertheless, on account of the theoretical and empirical importance of the sociodemographic and contextual variables of immigrant youths compared above, immigrant youths’ gender, age, number of siblings, standardized English and math scores, family composition, parental SES, reception mode, ethnic origin, school type, school location, and minority school status were all adjusted in the modeling procedures to preclude the possibility of confounding effects.

With respect to whether parental hostility, emotional rejection, and harsh family interactions with parents in early adolescence negatively predict immigrant youths’ successful college graduation in young adulthood, GLMM results revealed that parental hostility in early adolescence significantly and negatively predicted successful college graduation of immigrant youths in young adulthood (model 1, Table 2) (*b* = −0.224, *p* < 0.001), meaning that a unit increase in parental hostility resulted in the decreased odds of successful college graduation of immigrant youths by 20.1%. In model 2, parental emotional rejection significantly and negatively predicted successful college graduation of immigrant youths in young adulthood (*b* = −0.248, *p* < 0.001), explicating that each unit of increased parental emotional rejection induced the decreased odds of immigrant youths’ successful college graduation by 21.9%. Moreover, harsh family interactions with parents were significantly and strongly predictive of the reduced rates of successful college graduation of immigrant youths in young adulthood (*b* = −0.360, *p* < 0.001), meaning that each unit increased in harsh family interactions with parents in early adolescence led to the decreased odds of college graduation of immigrant youths in young adulthood by 30.2%. In fact, the significant negative effects of parental hostility, emotional rejection, and harsh family interactions with parents on successful college graduation of immigrant youths in young adulthood were sustained, even controlling for the various sociodemographic and contextual covariates mentioned above.

With respect to the effects of sociodemographic and contextual covariates on successful college graduation of immigrant youths in young adulthood, female immigrant youths, as compared to their male counterparts, significantly had the higher odds of successful college graduation across the three mixed-effects models by 23.9% to 25.7%, and older immigrant youths generally showed the reduced odds of completing college education by 7.4% to 7.9%. Moreover, 1.5- and 2-generation immigrant youths, compared to their 2.5-gneration peers, unanimously had the greater odds of college graduation by 46.1% and 61.4%, respectively. More importantly, immigrant youths from two-parent families and those with higher parental SES were more likely to obtain a college degree in adulthood than their counterparts from non-two-parent and economically poorer families, with the increased odds of 15.9% to 17.4% for the former and approximately 51% for the latter. For school-level variables, immigrant youths attending public schools significantly had the lower odds of successful college gradation in young adulthood by around 13% as compared to their counterparts who attended private schools. Furthermore, compared to their counterparts who attended suburban schools, immigrant youths studying in inner-city schools significantly had the lower odds of college graduation by 35.1% to 37%. Lastly, ethnic origin did not show significant effects on the educational success of immigrant youths in young adulthood, except for Middle East and African immigrant youths, as compared to their Northeast/Eastern Asian counterparts, who had the greater odds of completing college education by 2.145 to 2.307 times.

Table 3 shows the standardized effects of parental hostility, emotional rejection, and harsh family interactions with parents in early adolescence of immigrant youths on the academic aspirations of immigrant youths in late adolescence adjusted for their academic aspirations in early adolescence. Model 1 revealed that parental hostility significantly predicted lower academic aspirations of immigrant youths in late adolescence (*β* = −0.057, *p* < 0.001). In addition, parental emotional rejection predicted significantly lower academic aspirations of immigrant youths in late adolescence (*β* = −0.040, *p* < 0.05; model 2, Table 3). Furthermore, model 3 revealed that harsh family interactions with parents in early adolescence substantially and significantly predicted lower academic aspirations of immigrant youths in late adolescence (*β* = −0.060, *p* < 0.001). Notably, the academic aspirations of immigrant youths in early adolescence were robustly and significantly positively contributive to their academic aspirations in late adolescence across the three models (*β* = 0.464 to 0.468, *p* < 0.001). In terms of the effects of sociodemographic and contextual covariates, female and 1.5- and 2-generation immigrant youths significantly had higher academic aspirations in late adolescence than their male and 2.5-gernation counterparts. Moreover, immigrant youths from two-parent families and with higher parental SES significantly had higher academic aspirations in late adolescence than their counterparts from non-two-parent and economically poorer families. Furthermore, public school and inner-city school status significantly contributed to lower academic aspirations of immigrant youths in late adolescence at the school level.

Table 4 presents the results of the effects of parental hostility, emotional rejection, and harsh family interactions with parents in early adolescence of immigrant youths and academic aspirations of immigrant youths in late adolescence on successful college graduation of immigrant youths in young adulthood; GLMM procedures were conducted to examine their direct effects on successful college graduation of immigrant youths in young adulthood and to model constraints by setting new indirect paths to examine the mediated effects of immigrant youths’ academic aspirations in late adolescence on the relationships between parental hostility, emotional rejection, and harsh family interactions with parents in early adolescence of immigrant youths and successful college graduation of immigrant youths in young adulthood. Table 4 shows that both parental hostility in early adolescence and academic aspirations of immigrant youths in late adolescence significantly predicted successful college graduation in young adulthood (model 1; *b* = −0.191 and 0.814, *p* < 0.001); for each unit of increase in parental hostility, the odds of immigrant youths’ successful college graduation decreased by 17.4%, and for each unit of increase in academic aspirations of immigrant youths in late adolescence, the odds of successful college graduation increased by 2.257 times. Moreover, model 2 revealed that academic aspirations of immigrant youths in late adolescence significantly and positively predicted their successful college graduation (*b* = 0.807, *p* < 0.001), whereas parental emotional rejection in early adolescence significantly and negatively predicted successful college graduation of immigrant youths (*b* = −0.181, *p* < 0.01); for each unit of increase in academic aspirations of immigrant youths in late adolescence, the odds of educational success of immigrant youths in young adulthood increased by 2.242 times, and for each unit of increase in parental emotional rejection, the odds of immigrant youths’ successful college graduation in young adulthood decreased by 16.6%. In model 3, both the academic aspirations of immigrant youths in late adolescence and harsh family interactions with parents in early adolescence of immigrant youths significantly predicted successful college graduation of immigrant youths in young adulthood (*b* = 0.800 and −0.289, *p* < 0.001); for each unit of increase in academic aspirations of immigrant youths in late adolescence, the odds of successful college gladiation in young adulthood increased by 2.226 times, whereas for each unit of increase in harsh family interactions with parents, the odds of immigrant youths successfully obtaining a college degree in young adulthood decreased by 25.1%.

Table 5 presents the indirect and total indirect effects of immigrant youths’ academic aspirations in late adolescence on the relationships between parental hostility, emotional rejection, and harsh family interactions with parents in early adolescence and their successful college graduation in young adulthood. Indirect effects indicate the direct mediated effects of immigrant youths’ academic aspirations, and total indirect effects denote the combined indirect and direct effects of the mediator of immigrant youths’ academic aspirations in late adolescence and the predictors of parental hostility, emotional rejection, and harsh family interactions with parents in early adolescence of immigrant youths. In the mediational modeling procedures, all the sociodemographic and contextual covariates at the individual and school levels controlled for in the prior GLMM procedures were also adjusted. Academic aspirations of immigrant youths in late adolescence significantly mediated the effect of parental hostility on successful college graduation of immigrant youths in young adulthood, with an indirect effect (*β*_ind_ = −0.156, *p* < 0.001) and total indirect effect (*β*_totind_ = −0.156, *p* < 0.001). In addition, academic aspirations of immigrant youths in late adolescence significantly mediated the effect of parental emotional rejection on college graduation of immigrant youths in young adulthood, with an indirect effect (*β*_ind_ = −0.146, *p* < 0.01) and total indirect effect (*β*_totind_ = −0.156, *p* < 0.001). Lastly, it was found that immigrant youths’ academic aspirations in late adolescence significantly mediated the effect of harsh family interactions with parents in early adolescence of immigrant youths on their successful college graduation in young adulthood, with an indirect effect (*β*_ind_ = −0.231, *p* < 0.001) and total indirect effect (*β*_totind_ = −0.364, *p* < 0.001).

## 5. Discussion

This study represents a first attempt to investigate how harsh family experiences of immigrant youths in early adolescence adversely impact their educational achievement in young adulthood through the mediation of academic aspirations of immigrant youths in late adolescence, which is of research importance, due to the disadvantaged social conditions and cultural differences of immigrant families [13,49]. Some scholars have posited that harsh parental discipline and strict parenting are common and even regarded as “normative” in immigrant families and thought to be less harmful on the development of immigrant youths [9,11]. However, this study revealed that parental hostility, emotional rejection, and harsh family interactions with parents experienced by immigrant youths in early adolescence had significant negative effects on positive educational development in terms of development of academic aspirations in late adolescence and educational success in young adulthood across different ethnic origins. It is noted that parental hostility and emotional rejection—or their aggregate as harsh family interactions with parents—are common in many disadvantaged families, especially those of immigrant background [5,11]. This is in alignment with recent research suggesting that parental hostility and emotional indifference and rejection are common forms of parental maltreatment experienced by children and youths in disadvantaged and immigrant families [6,9,10,19]. Because minor forms of parental maltreatment are difficult to define and identify, they are easily overlooked by policy makers, educators, human service practitioners, and researchers. Nevertheless, limited relevant research recently revealed that these minor forms of parental maltreatment are far less negligible than severe parental maltreatment types, according to their frequency, chronicity, and incessancy [6,12]. The results reported in this study support the hypothesis that parental hostility, emotional rejection, and harsh family interactions with parents in early adolescence of immigrant youths significantly impair the likelihood of immigrant youths’ successful college graduation in young adulthood, giving profound implications for their later-life transformation and social mobility [10,32].

Moreover, harsh family interactions with parents, the aggregate of parental hostility and emotional rejection, were found to more strongly compromise both successful college graduation of immigrant youths in young adulthood and their development of academic aspirations in late adolescence than the single minor form of parental hostility and emotional rejection, revealing the accumulative detriment of minor forms of parental maltreatment with respect to the positive development of immigrant youths. This is congruous with the claim by some researchers that concurrent exposure to multiple types of harsh parental discipline and maltreatment could result in destructive impacts on youth development [5,36]. Therefore, effective implementation of early identification initiatives to help prevent common minor forms of parental maltreatment is an important strategy to help youths from being adversely affected in the long run. Collectively, educational and human service practitioners, as well as policy makers, should attend to early identification and timely family supports for immigrant families in order to support immigrant parents in delivering effective family socialization processes [11,57]. This is important because harsh family socialization experiences of immigrant youths in early years have been found here profoundly and adversely affect their educational success in young adulthood. Hence, timely and responsive family supports and programs targeted at immigrant families and their children are needed.

In addition, the results of this study show that parental hostility, emotional rejection, and harsh family interactions with parents significantly impaired the development of academic aspirations of immigrant youths in late adolescence, which, in turn, mediated the relationships between their experiences of harsh family socialization in early adolescence and successful college graduation in young adulthood. Similarly, recent research pointed out that the “future self” is an extension of the “current self” [41]. Hence, consistent with the self-system theory of motivational development, harshly disciplined and maltreated youths generally feel incapable, inferior, and worthless; as a result of their negative self-cognition, they may easily abandon their future hope for educational and social success [5]. Nevertheless, as the general self-identity and cognition of youths are socially constructed and cultivated, educational and human service practitioners should help immigrant youths with undermined self-concept and future hope develop a sense of positive self and establish aspirations for future success. This is useful because constructive future aspirations of youths “draws on the motivational properties of projecting the self onto the future….that future thinking regulates present behavior have been supported by empirical findings indicating its effect on each of multiple behaviors, including adolescents’ academic achievement [17] (p. 198)”. Therefore, future research should investigate the underlying process whereby harsh parental discipline and maltreatment may first enervate the intrapersonal self and future hope of youths that then become crucial cognitive and mental agents influencing their long-term educational, behavioral, psychological, and social development.

On the other hand, some sociodemographic and contextual covariates of immigrant youths were found to be consistently and significantly predictive of both immigrant youths’ successful college graduation in young adulthood and academic aspirations in late adolescence. Specifically, female and 1.5- and 2-generation immigrant youths showed higher rates of successful college graduation and academic aspirations than their male and 2.5-generation counterparts. Prior immigration research reported that female and less assimilated immigrant youths have higher achievement drive, which critically propels their higher educational and social attainments [42,46], as confirmed in the current study by their higher odds of college graduation and higher academic aspirations. Moreover, two-parent family status and parental SES were persistently and significantly found predictive of higher rates of college graduation and academic aspirations among immigrant youths, highlighting the importance of two-parent families and resourceful family socialization in immigrant youths’ educational and cognitive development [11,43]. Furthermore, immigrant youths who attended public and inner-city schools were found to have lower rates of successful college graduation and undermined academic aspirations than their counterparts who attended private and suburban schools. Some researchers reported that public and inner-city schools have more pronounced “oppositional cultures” and poorer teaching quality, which may hinder the learning and educational development of immigrant youths [32,46]. Thus, educational innovations to improve the learning environments of immigrant youths in these disadvantaged schools are urgently needed.

## 6. Conclusions

In summary, the current study reveals and supports a clear message that even minor forms of parental maltreatment and parental harshness in the early years of immigrant youths, such as parental hostility, emotional rejection, and harsh family interactions with parents in aggregate, may adversely affect immigrant youths’ educational success in adulthood; this significant relationship was persisted, even after adjusting for various important sociodemographic and contextual confounding covariates of immigrant youths, including gender, age, family characteristics, immigration generation, standardized test scores, ethnicity, school type, and locality at the individual and school levels. In addition, academic aspirations of youths were found to mediate the effects of parental hostility, emotional rejection, and harsh family interactions with parents in the early adolescence of immigrant youths on their later successful college graduation in adulthood. However, the current study is subject to several limitations that should be addressed in future investigations. First, although the current study included a large and representative sample of immigrant youths sampled from the two immigrant-receiving regions in the United States, these immigrant youths cannot represent the whole population of immigrant youths in other parts of the United States. Secondly, other serious types of parental maltreatment, e.g., sexual and physical abuse, and some important intrapersonal constructs of immigrant youths, such as self-esteem and life meaning, that may mediate the studied relationships, were not examined. Therefore, future research should investigate the accumulative effects of different maltreatment experiences of immigrant youths in relation to their educational achievement through the development of multiple cognitive and intrapersonal mediators. Thirdly, CILS data were collected twenty years ago and may not reflect the recent political economic, societal, and immigration policy changes in Western immigrant-receiving countries that are profoundly related to educational development of immigrant youths [58,59]. Therefore, more updated data and immigration research are needed to further confirm the studied relationships found between harsh family socialization experiences of immigrant youths and their educational development. Moreover, as limited by the data structure of the CILS, in the current study, we only employed single or a few items to measure parental hostility, emotional rejections, and harsh family interactions with parents, which is a major limitation of the current study, although some existing research supports the validity of using this measurement approach tapping on the minor forms of parental maltreatment [18,27]. Last but not least, because the CILS data do not include non-Hispanic White youths, who distinctively differ from immigrant youths culturally, economically, and socially, verification of cross-population validity and comparison by putting non-Hispanic White youths as reference was impossible in this study.

In general, helping immigrant families to develop and cultivate nurturant and supportive parent–child relationships and enhancing immigrant youths’ establishment of positive academic aspirations in early years are pivotal for their educational success in adulthood, all of which are important determinants of their well-being and health in later life trajectories. Therefore, policy maker, educators, and human service professionals should pay more attention to these negative family experiences of immigrant youths when designing policy, programming, and interventions. In conclusion, we believe that if timely and adequate interventions and service supports are provided to harshly disciplined and maltreated immigrant youths and their families, these youths can thrive academically and socially.

## Figures and Tables

**Table 1 ijerph-20-00873-t001:** Sociodemographic characteristic of immigrant youth participants in wave 1 of the CILS; *N* = 3344.

	Mean (%)	SD	Range
Gender			
Female			
Male	0.541		0, 1
Age	0.459		0, 1
Siblings	14.234	0.863	12–18
Generation	1.797	1.463	0–8
1.5 Generation			
2 Generation	0.469		0, 1
2.5 Generation	0.413		0, 1
Standardized English Scores	0.119		0, 1
Standardized Math Scores	700.762	56.184	0–857
Family Composition	669.111	58.792	0–830
Two-parent Family			
Other Family	0.758		0, 1
Parental SES	0.242		0, 1
Incorporation Mode	34.253	13.043	13–88
Positive/Neutral			
Negative	0.729		0, 1
Ethnic Background	0.271		0, 1
Cuban			
Mexican	0.249		0, 1
Caribbean	0.124		0, 1
Central/South American	0.109		0, 1
Southeast Asian	0.166		0, 1
Northeast/East Asian	0.289		0, 1
Middle Eastern/African	0.023		0, 1
European	0.019		0, 1

Note: Parental SES was calculated according to the Duncan Socioeconomic Index.

**Table 2 ijerph-20-00873-t002:** Generalized linear mixed modeling predicting the effects of parental hostility, emotional rejection, and harsh family interactions with parents in early adolescence of immigrant youths on their successful college graduation in young adulthood.

	Model 1		Model 2		Model 3	
	b	OR	b	OR	b	OR
Individual Level						
Female	0.228 ***	1.257	0.215 ***	1.239	0.226 ***	1.254
Age	−0.082 *	0.921	−0.077 *	0.925	−0.077 *	0.926
Siblings	−0.051	0.950	−0.035	0.966	−0.041	0.960
1.5 Generation	0.479 ***	1.614	0.453 **	1.573	0.476 **	1.610
2 Generation	0.391 **	1.478	0.379 **	1.461	0.391 **	1.478
Standardized Math Score	0.034	1.035	0.029	1.029	0.029	1.029
Standardized English Score	−0.078	0.925	−0.075	0.927	−0.077	0.926
Two-parent Family	0.160 **	1.174	0.147 **	1.159	0.149 **	1.161
Parental SES	0.413 ***	1.512	0.417 ***	1.517	0.412 ***	1.510
Negative Reception	−0.086	0.918	−0.101 *	0.904	−0.095	0.910
Cuban	0.300	1.350	0.373	1.452	0.310	1.364
Mexican	0.396	1.486	0.458	1.581	0.406	1.500
Caribbean	0.411	1.509	0.479 *	1.615	0.421	1.523
Central and South American	0.190	1.210	0.250	1.284	0.198	1.219
Southeast Asian	0.321	1.378	0.362	1.436	0.318	1.374
Middle Eastern/African	0.782 *	2.186	0.836 *	2.307	0.763 *	2.145
European	0.161	1.174	0.222	1.248	0.177	1.193
Parental Hostility	−0.224 ***	0.799				
Parental Emotional Rejection			−0.248 ***	0.781		
Harsh Family Interactions					−0.360 ***	0.698
School Level						
Public School	−0.139 *	0.870	−0.133 *	0.875	−0.137 *	0.871
Minority School	0.021	1.021	−0.019	0.981	0.000	1.000
Inner-city School	−0.461 ***	0.630	−0.431 **	0.649	0.452 ***	0.636
Intercept τ00	0.066		0.061		0.062	
$R_{within}^{2}$	0.100		0.100		0.107	
$R_{between}^{2}$	0.573		0.551		0.574	

Note: b is the regression effects, and OR represents the odds ratios obtained by exponentiating the regression betas. * *p* < 0.05; ** *p* < 0.01; *** *p* < 0.001.

**Table 3 ijerph-20-00873-t003:** Generalized linear mixed modeling predicting the effects of parental hostility, emotional rejection, and harsh family interactions with parents in early adolescence of immigrant youths on academic aspirations of immigrant youths in late adolescence.

	Model 1		Model 2		Model 3	
	β	*t*	β	*t*	β	*t*
Individual level						
Female	0.088 ***	5.485	0.085 ***	5.245	0.088 ***	5.405
Age	−0.074 **	−3.278	−0.073 **	−3.185	−0.072 **	−3.204
Siblings	−0.011	−0.0863	−0.008	−0.604	−0.009	−0.669
1.5 Generation	0.080 **	3.370	0.076 **	3.264	0.080 **	3.370
2 Generation	0.057 *	2.082	0.055 *	2.018	0.057 *	2.086
Standardized Math Score	0.049 *	2.450	0.048 *	2.336	0.048 *	2.383
Standardized English Score	−0.031	−1.667	−0.030	−1.664	−0.030	−1.676
Intact Family	0.035 *	2.327	0.0033 *	2.244	0.033 *	2.175
Parental SES	0.076 ***	4.633	0.077 ***	4.782	0.076 ***	4.670
Negative Reception	−0.037	−1.544	−0.040	−1.671	−0.039	−1.638
Cuban	0.064	1.318	0.069	1.390	0.067	1.365
Mexican	0.050	1.204	0.053	1.250	0.052	1.265
Caribbean	0.047	1.388	0.050	1.465	0.049	1.440
Central and South American	0.044	0.957	0.048	1.021	0.046	1.003
Southeast Asian	0.080	1.457	0.082	1.464	0.082	1.475
Middle Eastern/African	0.015	0.738	0.016	0.790	0.015	0.741
European	0.016	0.667	0.017	0.706	0.017	0.706
Wave-1 Academic Aspirations	0.468 ***	23.421	0.466 ***	23.718	0.464 ***	23.559
Parental Hostility	−0.057 ***	−3.490				
Parental Emotional Rejection			−0.040 *	−2.304		
Harsh Family Interactions					−0.060 ***	−3.930
School Level						
Public School	−0.290 *	−2.187	−0.287 *	−2.127	−0.293 *	−2.181
Minority School	0.089	0.413	0.016	0.069	0.046	0.213
Inner-city School	−0.558 **	−3.383	−0.558 **	−3.171	−0.568 **	−3.379
Intercept τ00						
$R_{within}^{2}$	0.292		0.292		0.293	
$R_{between}^{2}$	0.492		0.463		0.491	

* *p* < 0.05; ** *p* < 0.01; *** *p* < 0.001.

**Table 4 ijerph-20-00873-t004:** Generalized linear mixed modeling predicting the effects of parental hostility, emotional rejection, and harsh family interactions with parents in early adolescence of immigrant youths and academic aspirations of immigrant youths in late adolescence on their successful college graduation in young adulthood.

	Model 1		Model 2		Model 3	
	b	OR	b	OR	b	OR
Individual Level						
Female	0.151 ***	1.163	0.139 ***	1.149	0.150 ***	1.162
Age	−0.022	0.978	−0.020	0.981	−0.019	0.981
Siblings	−0.029	0.972	0.019	0.981	−0.021	0.979
1.5 Generation	0.372 **	1.451	0.346 *	1.413	0.367 **	1.443
2 Generation	0.304 *	1.356	0.288 *	1.333	0.300 *	1.350
Standardized Math Score	0.001	1.001	0.000	1.000	−0.001	0.999
Standardized English Score	−0.058	0.944	−0.056	0.946	−0.057	0.944
Two-parent family	0.144 **	1.155	0.136 **	1.145	0.137 **	1.147
Parental SES	0.350 ***	1.419	0.353 ***	1.424	0.350 ***	1.419
Negative Reception	−0.064	0.938	−0.074	0.928	0.071	0.932
Cuban	0.273	1.314	0.335	1.398	0.285	1.330
Mexican	0.382	1.465	0.436	1.547	0.395	1.484
Caribbean	0.405	1.499	0.464	1.591	0.417	1.518
Central and South American	0.173	1.189	0.222	1.248	0.182	1.199
Southeast Asian	0.303	1.353	0.340	1.404	0.303	1.354
Middle Eastern/African	0.758	2.133	0.806	2.240	0.747	2.111
European	0.204	1.226	0.255	1.291	0.216	1.241
Wave-2 Future Academic Aspirations	0.814 ***	2.257	0.807 ***	2.242	0.800 ***	2.226
Parental Hostility	−0.191 ***	0.826				
Parental Emotional Rejection			−0.181 **	0.834		
Harsh Family Interactions					−0.289 ***	0.749
School Level						
Public School	−0.114 *	0.892	−0.108 *	0.897	−0.113 *	0.893
Minority School	0.003	1.003	−0.027	0.973	−0.015	0.985
Inner-city School	−0.352 **	0.703	−0.320 *	0.726	−0.341 **	0.711
Intercept τ00	0.047		0.044		0.045	
$R_{within}^{2}$	0.218		0.215		0.220	
$R_{between}^{2}$	0.528		0.494		0.520	

* *p* < 0.05; ** *p* < 0.01; *** *p* < 0.001.

**Table 5 ijerph-20-00873-t005:** Mediated effects of academic aspirations of immigrant youths in late adolescence on the relationships between parental hostility, emotional rejection, and harsh family interactions with parents in early adolescence of immigrant youths and their successful college graduation in young adulthood.

Path through Academic Aspirations		Indirect Effect	z-Value	Total Indirect Effect	z-Value
1.	Parental hostility ≥ college graduation	−0.156 ***	−3.652	−0.229 ***	−5.029
2.	Parental emotional rejection ≥ college graduation	−0.146 **	−2.567	−0.252 ***	−4.188
3.	Harsh family interactions ≥ college graduation	−0.231 ***	−3.713	−0.364 ***	−5.219

Note: In modeling the indirect and total indirect effects, all the sociodemographic and contextual covariates presented in models 1 to 3 in Table 4 were adjusted for. * *p*< 0.05; ** *p* < 0.01; *** *p* < 0.001.

## Data Availability

The data of CILS is available at The Center for Migration and Development in Princeton University, please visit https://cmd.princeton.edu/publications/data-archives/cils.

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
