# Peer review of "Minor Forms of Parental Maltreatment and Educational Achievement of Immigrant Youths in Young Adulthood: A Longitudinal Study"

_ijerph, 2023, doi:10.3390/ijerph20010873_

Round 1

Reviewer 1 Report

The present manuscript aims to explore the relation between parental hostility and emotional rejection on academic aspiration among Immigrant students. Parental hostility seems to be captured though (i) “in the process of family socialization and communication with parents were full of clash and hostility from their parents”. Parental emotional rejection was captured though two items: (i) whether their parents did not like him/her much; and (ii) their parents were not interested in the way he or she was. Overall, the results showed the negative impact of on academic aspiration. The manuscript should be adequately rationalized within the text of the manuscript connecting previous classical and recent research with the focus of the study, specially about parenting.

An important question that should be considered are the measures of parental hostility and emotional rejection. Authors should add more details about the items of parental hostility. It is difficult to find the specific measure, so it should be included the items. The measures seem to be like those about parental responsiveness (lower parental responsiveness). It seems that parental hostility does not include sexual or physical abuse, but the items should be included specifically.

(1)    Introduction

Within the text of the manuscript, authors should characterize the parental practices of parental hostility and emotional rejection within the model of family socialization based on two dimensions (i.e., responsiveness and demandingness). The dimension of responsiveness represents parental love, approval, acceptance, and support (Gimenez-Serrano et al., 2022). Parental practices of rejection and hostility are characterized by lower levels of responsiveness (Fuentes et al., 2015; Garcia et al., 2020). Overall, results revealed the negative impact of parenting characterized by poor responsiveness (Garcia & Gracia, 2009; Yeung, 2021).

Additionally, the difficulties of immigrant students should be considered more detailed, being immigrant might be related to some problems such as less academic engagement compared to native students (Veiga et al. 2021). The same is true for children in risk neighborhood who had more problems than those from middle-class (Sandoval et al., 2022).

References

Fuentes, M. C., Alarcón, A., Gracia, E., & García, F. (2015). School adjustment among Spanish adolescents: Influence of parental socialization. Cultura y Educación, 27, 1-32. https://doi.org/10.1080/11356405.2015.1006847

García, F., & Gracia, E. (2009). Is always authoritative the optimum parenting style? Evidence from Spanish families. Adolescence, 44(173), 101-131. 

Garcia, O. F., Fuentes, M. C., Gracia, E., Serra, E., & Garcia, F. (2020). Parenting warmth and strictness across three generations: Parenting styles and psychosocial adjustment. International Journal of Environmental Research and Public Health, 17(7487), 1-18. https://doi.org/10.3390/ijerph17207487

Gimenez-Serrano, S., Alcaide, M., Reyes, M., Zacarés, J. J., & Celdrán, M. (2022). Beyond parenting socialization years: The relationship between parenting dimensions and grandparenting functioning. International Journal of Environmental Research and Public Health, 19(4528), 1-13. https://doi.org/10.3390/ijerph19084528

Sandoval-Obando, E., Alcaide, M., Salazar-Muñoz, M., Peña-Troncoso, S., Hernández-Mosqueira, C., & Gimenez-Serrano, S. (2022). Raising children in risk neighborhoods from Chile: Examining the relationship between parenting stress and parental adjustment. International Journal of Environmental Research and Public Health, 19(45), 1-14. https://doi.org/10.3390/ijerph19010045

Veiga, F. H., Festas, I., García, Ó. F., Oliveira, Í. M., Veiga, C. M., Martins, C., Covas, F., & Carvalho, N. A. (2021). Do students with immigrant and native parents perceive themselves as equally engaged in school during adolescence? Current Psychology. https://doi.org/10.1007/s12144-021-02480-2 

Yeung, J. K. (2021). Family processes, parenting practices, and psychosocial maturity of Chinese youths: A latent variable interaction and mediation analysis. International Journal of Environmental Research and Public Health, 18(4357), 1-15. https://doi.org/10.3390/ijerph18084357

(2)     Discussion

Parental socialization ends when the child arrives to the adult age. The results confirm other previous studies about the long-term impact of parenting with adult children (Candel et al., 2020). Additionally, the statistical power should be considered more detailed as strong point of the study (Faul et al., 2007; Perez et al., 1999).

References

Candel, O. (2022). The link between parenting behaviors and emerging adults’ relationship outcomes: The mediating role of relational entitlement. International Journal of Environmental Research and Public Health, 19(828), 1-12. https://doi.org/10.3390/ijerph19020828

Faul, F., Erdfelder, E., Lang, A. G., & Buchner, A. (2007). G*Power 3: A flexible statistical power analysis program for the social, behavioral, and biomedical sciences. Behavior Research Methods, 39, 175-191. https://doi.org/10.3758/BF03193146

Pérez, J. F. G., Navarro, D. F., & Llobell, J. P. (1999). Statistical power of Solomon design. Psicothema, 11, 431-436.

Author Response

Thank for reviewing our submitted manuscript titled “Harsh Family Interactions with Parents and Educational Achievement of Immigrant Youths in young adulthood: A Longitudinal Study”. We have revised the whole manuscript according to the comments of the reviewers and our responses are follows:

For Reviewer 1

1) The manuscript should be adequately rationalized within the text of the manuscript connecting previous classical and recent research with the focus of the study, specially about parenting.

Reply: Now, the theoretical framework of the manuscript has been thoroughly rewritten, which include “connecting previous classical and recent research with the focus of the study, specially about parenting.” For this, please refer the parts of ““Harsh family interactions with parents and educational achievement” and “The mediation of academic aspiration of immigrant youths”.

2) An important question that should be considered are the measures of parental hostility and emotional rejection. Authors should add more details about the items of parental hostility. It is difficult to find the specific measure, so it should be included the items. The measures seem to be like those about parental responsiveness (lower parental responsiveness). It seems that parental hostility does not include sexual or physical abuse, but the items should be included specifically.

Reply: Now parental hostility and emotional rejection as well as their aggregate as harsh family interaction with parents in relation to academic aspiration and college graduation of immigrant youths in young adulthood have been explained, constructed, and justified in details. For details, please refer to the parts of ““Harsh family interactions with parents and educational achievement” and “The mediation of academic aspiration of immigrant youths”. However, the current study aims to investigate how the minor forms of parental harsh discipline and maltreatment, such as parental hostility and emotional rejection as well as their aggregate as harsh family interaction with parents, which have been received less research attention compared to investigations in parental physical and sexual abuse. In fact, as the dataset of CILS was not planned to investigate family violence and child abuse but the life transformation of immigrant youths after immigration to the United States, hence it does not allow to include those serious child maltreatment constructs, which now has elaborated in the manuscript as its limitations.

3) Within the text of the manuscript, authors should characterize the parental practices of parental hostility and emotional rejection within the model of family socialization based on two dimensions (i.e., responsiveness and demandingness). The dimension of responsiveness represents parental love, approval, acceptance, and support (Gimenez-Serrano et al., 2022). Parental practices of rejection and hostility are characterized by lower levels of responsiveness (Fuentes et al., 2015; Garcia et al., 2020). Overall, results revealed the negative impact of parenting characterized by poor responsiveness (Garcia & Gracia, 2009; Yeung, 2021).

Reply: Now the framework of family socialization model has been put in the study of parental hostility, emotional rejection, and harsh family interactions with parents to explain why and how parental hostility, emotional rejection, and harsh family interactions with parents would negatively affect educational success and academic aspiration of immigrant youths, which are written:

“Correspondently, family transmission model evinces that parents are the main socialization figures responsible to provide care, support, resources, and social and cultural capitals necessary for cognitive and healthy development of their offspring [3]. However, harshly disciplined and maltreated youths even experienced by the minor forms of parental hostility, emotional rejection, or harsh family interactions with parents in family may signify their insufficiency in parental educational involvement and support as well as acquisition of necessary learning resources and guidance to help them overcome academic challenges for achieving educational success [11, 12], as a result posing them poorer educational development. This is especially important for immigrant youths as they are generally living under the conditions of family poverty and weak social supports and resources [31, 32]. Yet, although limited pertinent research has reported the harmful impacts of parental hostility and rejection on behavioral and psychological adjustments of children and youths, little is known about how these negative family socialization experiences may negatively affect educational development of immigrant youths. This is research-worthy from the model of family socialization that youths experiencing harsh discipline and maltreatment implies their receiving low parental responsiveness and support and high parental negativity [22, 33, 34], which are harmful to youth health development educationally and socially. In their meta-analysis, Khaleque [26] found that paternal hostility/aggression significantly associated with psychological maladjustment and negative personality dispositions of children across ethnicities, cultures, and geographical boundaries. Similarly, in their latest study, Backman, Laajasalo, Jokela and Aronen [23] found that parental hostility and lower maternal warmth were significantly predictive of psychopathic behaviors of offending adolescents. More recently, Lee and Mun [35] reported that parental rejection was positively related to cyberbullying behaviors of Korean children and youths of perpetration in a representative sample from the 2019 Korean Children and Youth Panel Survey (KCYPS), in which the study relationship was independent and sequentially mediated by children’s depression and smartphone addiction. Academically, Putnick, Bornstein, Lansford, Malone, Pastorelli, Skinner, Sorbring, Tapanya, Tirado, Zelli, Alampay, Al-Hassan, Bacchini, Bombi, Chang, Deater-Deckard, Di Giunta, Dodge and Oburu [8] found that parental rejection was longitudinally predictive of lower school scores among primary school children across nine countries. Moreover, Ryan, Jacob, Gross, Perron, Moore and Ferguson [20] investigated 732,828 youths born between 2000 and 2006 in Michigan public schools and found that those of child protective service involvement (CPS) exhibited significantly lower math and reading scores, grade retention, and receipt of special education than their peers without CPS involvement. More relevantly, Welsh, et al. [36] found that college students who experienced a history of maltreatment significantly exhibited lower first-semester GPA and poorer college adaptation.” .”(Lines 110-201 in the part of “Harsh family interactions with parents and educational achievement”)

“As academic aspiration of youths has been found to be shaped by the process of family socialization and experiences in early years [14, 15], it is believed in this study that the academic aspiration of immigrant youths in late adolescence not only will directly affect their educational achievement in young adulthood [12, 41, 42], but also may mediate the effects of parental hostility, emotional rejection, and harsh family interactions with parents on their later educational success in young adulthood [17, 38]. In a longitudinal study by Hentges and Wang [14], they found that harsh parenting in 7th grade was significantly related to students’ lower GPA in 11th grade through the mediation of compromised academic values in 8th grade. Moreover, Seginer and Mahajna [17] reported that perceived positive parenting significantly and positively predicated Muslim youths’ academic aspiration for higher education in 11th-grade that in turn mediated the relationship between parenting and academic performance. In her qualitative study, Morton [15] explored how maltreated experiences of foster youths might incur cognitive and behavioral difficulties in relation to their compromised educational achievement and she found that maltreated youths generally held lower academic aspiration that would seriously harm their success of college education. Therefore, in this study, we expect that immigrant youths’ experiences of parental hostility, emotional rejection, and harsh family interactions with parents in early adolescence may negatively impair their development of academic aspiration in late adolescence and successful college graduation in young adulthood, in which the development of immigrant youths’ academic aspiration in late adolescence would mediate the relationships of immigrant youths’ experiences of parental hostility, emotional rejection, and harsh family interactions with parents in early adolescence and their successful college graduation in young adulthood.”(Lines 529-551 in the part of “The mediation of academic aspiration of immigrant youths”)

4) Additionally, the difficulties of immigrant students should be considered more detailed, being immigrant might be related to some problems such as less academic engagement compared to native students (Veiga et al. 2021). The same is true for children in risk neighborhood who had more problems than those from middle-class (Sandoval et al., 2022).

Reply: Right, the current study has mentioned the disadvantaged family and contextual situations that negatively affect educational success of immigrant youths, which include:

“Conspicuously, immigrant youths compared to their non-immigrant counterparts are at greater risk of being harshly disciplined and maltreated by their parents due to lower parental education, disadvantaged social status, and cultural differences [11]. This is correspondent with what Romano, et al. [12] mentioned “socioeconomic disadvantage is important because such stressful conditions as poverty often compromise effective parenting practices (p. 433).” However, no longitudinal research today has been conducted to investigate the negative effects of parental hostility, emotional rejection, and their aggregate as general harsh family interactions with parents experienced by immigrant youths on their educational success in young adulthood. This is important as obtaining a college degree connotes immigrant youths a life chance of upward social mobility, which resonates with Portes, et al. [13] mentioned “(w)ithout the costly and time-consuming achievement of a university degree, such dreams are likely to remain beyond reach (p.1081).””(Lines 62-74 in the part of “Introduction”)

“Correspondently, family transmission model evinces that parents are the main socialization figures responsible to provide care, support, resources, and social and cultural capitals necessary for cognitive and healthy development of their offspring [3]. However, harshly disciplined and maltreated youths even experienced by the minor forms of parental hostility, emotional rejection, or harsh family interactions with parents in family may signify their insufficiency in parental educational involvement and support as well as acquisition of necessary learning resources and guidance to help them overcome academic challenges for achieving educational success [11, 12], as a result posing them poorer educational development. This is especially important for immigrant youths as they are generally living under the conditions of family poverty and weak social supports and resources [31, 32].” (Lines 110-175 in the part of “Harsh family interactions with parents and educational achievement”)

“Accordingly, as immigrant parents are generally of disadvantaged social status, economic deprivation, and cultural differences, they may be more prone of employing harsh discipline and unsupportive parenting [11, 27, 37]. Therefore, it is expected that experiencing parental hostility, emotional rejection, and/or harsh family interactions with parents in family is more common for immigrant youths, which is believed to detriment their educational achievement in adulthood, such as successful college graduation with a four-year undergraduate degree . As such, due to the more disadvantaged and difficult family socialization environment of immigrant youths encountered and the importance of educational success for their social mobility in adulthood as compared to their better-off local counterparts [38, 39], it is theoretically and empirically important to study how immigrant youths’ experiences of parental hostility, emotional rejection, and harsh family interactions with parents in early adolescence may adversely affect their later successful college graduation in young adulthood.”( Lines 203-215 in the part of “Harsh family interactions with parents and educational achievement”)

5) Parental socialization ends when the child arrives to the adult age. The results confirm other previous studies about the long-term impact of parenting with adult children (Candel et al., 2020). Additionally, the statistical power should be considered more detailed as strong point of the study (Faul et al., 2007; Perez et al., 1999).

Reply: Yes, and agree. For power test, it is mainly aimed to ensure a sample size that is large enough for the probability of rejecting the null hypothesis when, in fact, it is false and avoiding a Type II error in order to truly confirm the alternative hypothesis when the null hypothesis is false. In fact, when the sample size increases the power ensues to increase, especially for the samples drawn from representative sampling procedures, like the representative sample used in the current study (N=3344). Even using Z-test of power analysis by setting alpha level at 0.01 and 1-beta error probability level at 0.99, the strictest standard of power test for sample size, the result shows that N=2195 is well enough to ensure the power of rejecting the null hypothesis and confirming the alternative hypothesis. Hence, as it is a well-known fact regarding the assurance of power representative samples, I think it is less relevant to especially highlight the relationship between power and sample size in the current study.

As our responses to Reviewer 2 are also relevant to what Reviewer 1 concerns, hence we also enclosed our responses to Reviewer 2 here:

For reviewer 2

1) The manuscript is generally well-written and the line of thought is easy to follow. However, my most important concern is the operationalization of parental hostility measures, which make me much more cautious than the authors to draw firm conclusions from this report.  

Reply: Thank you, for the operationalization of parental hostility, emotional rejection, and their aggregate as harsh family interactions with parents, which are now thoroughly revised and clearly presented to justify its meanings, reasons, influences and limitations by theories, past empirical research, and rational considerations based on the available information provided by the Children of Immigrants Longitudinal Study (CILS), where the current study obtain its dataset. For details of the revisions, please refer to the parts of “Harsh family interactions with parents and educational achievement” and “The mediation of academic aspiration of immigrant youths”, and “The present study”, as well as “Conclusion (Lines: 1709-1717)”. In fact, we now also have rewritten the presentations of the effects of parental hostility, emotional rejection, and their aggregate as harsh family interactions with parents on educational development of immigrant youths in the parts of “Discussion” and “Conclusion” to make the contents and presentations more accurate and valid, which include:

“This study is the first attempt to investigate how harsh family experiences of immigrant youths in early adolescence adversely impact their educational achievement in young adulthood through the mediation of academic aspiration of immigrant youths in late adolescence, which is of research importance due to disadvantaged social conditions and cultural differences of immigrant families [13, 49]. Some scholars reckoned that harsh parental discipline and strict parenting are common and even regarded as “normative” in immigrant families, which are thought less harmful to development of immigrant youths [9, 11]. However, this study found that parental hostility, emotional rejection, and harsh family interactions with parents experienced by immigrant youths in early adolescence occasioned long-lasting detriments to positive educational development of immigrant youths in terms of development of academic aspiration in late adolescence and educational success in young adulthood across different ethnic origins. It is noted that parental hostility and emotional rejection, or their aggregate as harsh family interactions with parents, are common in many disadvantaged families, especially those of immigrant background [5, 11]. This accords with some recent research suggesting that parental hostility and emotional indifference and rejection are common forms of parental maltreatment experienced by children and youths in disadvantaged and immigrant families [6, 9, 10, 19]. In fact, as minor forms of parental maltreatment are hard to define and identify, which makes it easily overlooked by policy makers, teachers, human service practitioners, and researchers. Nevertheless, limited relevant research recently found that these minor forms of parental maltreatment are far less from negligibility than those severe parental maltreatment types due to their frequency, chronicity, and incessancy [6, 12]. It is supported in this study that parental hostility, emotional rejection, and harsh family interactions with parents in early adolescence significantly impaired the likelihood of immigrant youths’ successful college graduation in young adulthood, bearing profound implications for their later life transformation and social mobility [10, 32].” (Lines 1501-1571 in the part of “Discussion”)

“Moreover, as limited by the data structure of CILS, the current study only employed single or few items to measure parental hostility, emotional rejections, and harsh family interactions with parents, which is a major limitation of the current study although some existing research supports the validity of using this measurement approach to tap on minor forms of parental maltreatment [18, 27].” (Lines 1711-1759 in the part of “Conclusion”)

2) For the hypotheses, I wonder what the added value is of looking at the two 'subscales' of the 'harsh family interactions' variable separately, if the final hypotheses concern the composite of these two.

Reply: The current study investigates how parental hostility, emotional rejection, and their aggregate as harsh family interactions with parents may negatively affect successful college graduation of immigrant youths separately and jointly is necessary. This is because they can happen independently and/or concurrently. Hence, compare and investigate the independent and joint effects of parental hostility and emotional rejection on educational development of immigrant youths are academically and socially important, which accords with double-whammy thesis that is to anticipate higher exposure to multiple types of harsh parental discipline and maltreatment would impair youth development more serious although different types of  multiple types of harsh parental discipline and maltreatment may sometimes happen separately. For this, we now have rewritten the relevant presentations to:

“Nevertheless, parental hostility and emotional rejection or aggregated as a general form of harsh family interactions with parents in family have not received adequate research attention, which are mainly due to their differences to physical and sexual abuse and are regarded as a minor form of parental maltreatment or just harsh discipline [23, 24]. This corresponds to what Allan, et al. [25] mentioned “parental hostility, as opposed to physical or sexual abuse and neglect, has not received much attention. …Abuse generally consists of physical or sexual brutality whereas hostility generally consists of verbal and psychological abusiveness such as anger or antagonism (p.169).” In addition, Khaleque [26] stated that “(p)arental (emotional) rejection, on the other hand, refers to the withdrawal or lack of parental warmth, affection, care, comfort, nurturance, support, or love toward their children (p.977).” Hence, it is justified to conceptualize parental hostility in this study as the manifestation of conflicting and clashing relationships from parents toward their youth children [25, 27], and parental emotional rejection as parental display of absent interest, warmth, love, and nurturance toward their youth children in an aversive and unapproved way [28, 29], and harsh family interactions with parents as the aggregate of these conflicting, clashing, rejecting, apathetic, and disapproving processes of parent-youth interactions [25, 28, 30].” (Lines 92-108 in the part of “Harsh family interactions with parents and educational achievement”)

“Yet, although limited pertinent research has reported the harmful impacts of parental hostility and rejection on behavioral and psychological adjustments of children and youths, little is known about how these negative family socialization experiences may negatively affect educational development of immigrant youths. This is research-worthy from the model of family socialization that youths experiencing harsh discipline and maltreatment implies their receiving low parental responsiveness and support and high parental negativity [22, 33, 34], which are harmful to youth health development educationally and socially.” (Lines 175-182 in the part of “Harsh family interactions with parents and educational achievement”)

“Accordingly, as immigrant parents are generally of disadvantaged social status, economic deprivation, and cultural differences, they may be more prone of employing harsh discipline and unsupportive parenting [11, 27, 37]. Therefore, it is expected that experiencing parental hostility, emotional rejection, and/or harsh family interactions with parents in family is more common for immigrant youths, which is believed to detriment their educational achievement in adulthood, such as successful college graduation with a four-year undergraduate degree. As such, due to the more disadvantaged and difficult family socialization environment of immigrant youths encountered and the importance of educational success for their social mobility in adulthood as compared to their better-off local counterparts [38, 39], it is theoretically and empirically important to study how immigrant youths’ experiences of parental hostility, emotional rejection, and harsh family interactions with parents in early adolescence may adversely affect their later successful college graduation in young adulthood. In addition, immigrant youths’ experiences of parental hostility and emotional rejection may happen independently or jointly by the form of harsh family interactions with parents in family, which is congruous with the double-whammy claim that higher exposure to multiple types of harsh parental discipline and maltreatment concurrently would impair youth development further adversely [36]. Thereby, it is justifiable to examine the respective and combined effects of parental hostility, emotional rejection, and harsh family interactions with parents on educational achievement of immigrant youths in adulthood.” (Lines 203-511 in the part of “Harsh family interactions with parents and educational achievement”)

“Moreover, as limited by the data structure of CILS, the current study only employed single or few items to measure parental hostility, emotional rejections, and harsh family interactions with parents, which is a major limitation of the current study although some existing research supports the validity of using this measurement approach to tap on minor forms of parental maltreatment [18, 27].” (Lines 1699-1704 in the part of “Conclusion”)

Besides, the whole manuscript has been thoroughly revised to make the conceptualization and operationalization of parental hostility, emotional rejection, and harsh family interactions with parents more justifiable and understandable, especially in the part of “Measures (Lines 981-1030)”.

In sum, we would like to confirm in the current study that the minor forms of parental hostility and emotional rejection would adversely impact educational development of immigrant youths longitudinally, and their aggregate as harsh family interactions with parents would be more harmful on educational development of immigrant youths. Accordingly, our results generally support this double-whammy postulation.

3) The variables mentioned in lines 147-150 (previous paragraph) are not mentioned as covariates under Present study, even though the analyses do include these variables as covariates.

Reply: The variables have been stated as covariates, in which their impacts on educational motivation and success of immigrant youths have further elaborated more clearly now:

“Hence, in this study, we incorporate the contextual effects of family composition, parental SES, and mode of reception at individual level and school type, school minority status, and school location at school level as covariates when examining the study relationships of parental hostility, emotional rejection, and harsh family interactions with parents in contribution to academic aspiration of immigrant youths and their successful college graduation in young adulthood.” (Lines 808-812 in the part of “Contextual influences and educational achievement of immigrant youths”)

“Moreover, this study adjusted for gender, age, number of siblings, generation status, standardized English and math test scores and ethnic origins of immigrant youths for precluding sociodemographic confounding effects. Immigration research reported that female immigrant youths generally performed better academically than their male counterparts [47], and older aged immigrant youths had more difficulties in school adjustment than their younger counterparts [32]. Besides, immigrant youths of more siblings connote their family supports and care are diluted, which may compromise their school performance [32, 42]. For immigrant generation status, earlier generations of immigrant youths are reported to have higher educational motivation and performance than their later generations, albeit their economic and social disadvantages, which is called as “generation mystery” [48]. Additionally, ethnic origins of immigrant youths are found to have different effects on their educational achievement, in which immigrant youths of Northeast/ Eastern Asian origin showed higher educational success and Mexican and Caribbean immigrant youths manifestly fell behind academically while other ethnic groups were in-between [42, 49]. Therefore, immigrant youths in this study are classified into eight major ethnic groups with reference to their cultural and geographic adjacency [32, 49], which include Mexican, Caribbean, Cuban, Central and Southern American, Southeast Asian, Northeast/ Eastern Asian, Middle Eastern and African, European origins.” (Lines 919-936 in the part of “The present study”)

And their effects on academic aspiration of immigrant youths and their successful college graduation are also more clearly reported:

“For the effects of sociodemographic and contextual covariates on successful college graduation of immigrant youths in young adulthood, female immigrant youths, as compared to their male counterparts, had significantly the higher odds of successful college graduation across the three mixed-effects models by 23.9% to 25.7%, and older aged immigrant youths generally showed the reduced odds of completing college education by 7.4% to 7.9%. Moreover, 1.5- and 2-generation immigrant youths, when comparing to their 2.5-gneration peers, unanimously had the greater odds of college graduation by 46.1% to 61.4%. More important, immigrant youths from intact family and of better parental SES were more likely to obtain a college degree than their counterparts from broken and economically poorer family by the higher odds of 15.9% to 17.4% for the former and around 51% for the latter. For school-level variables, immigrant youths attending public secondary schools significantly had the lower odds of successful college gradation by around 13% as compared to their counterparts from private schools. Furthermore, as compared to their counterparts from rural and suburban schools, immigrant youths studying in inner-city schools significantly had the lower odds of college graduation by 35.1% to 37%. Lastly, ethnic origins did not show significant effects on educational success of immigrant youths in young adulthood, except Middle-East and African immigrant youths, as compared to their Northeast/ Eastern Asian counterparts, had the greater odds of completing college education by 2.145 to 2.307 times.” (Lines 1316-1334 in the part of “Results”).

“Examining the effects of sociodemographic and contextual covariates, female and 1.5- and 2-generation immigrant youths significantly had higher academic aspiration in late adolescence than their male and 2.5-gernation counterparts. Besides, immigrant youths from intact family and of better parental SES significantly had higher academic aspiration in late adolescence than their counterparts from broken and economically poorer family. What’s more, public school and inner-city school statuses significantly contributed to lower academic aspiration of immigrant youths in late adolescence at school level.” (Lines 1347-1370 in the part of “Results”).

4) The original sample comprised of 5262 participants, this report included only 3344. Can the authors expand a bit more about the representativeness of this sample? E.g., is there any information on differences/reasons why other participants dropped out and how might this affect this study’s results?

Reply: The current study employs the sample of 3344 immigrant youths who had provided valid information regarding the study variables (parental hostility, emotional rejection, and their aggregate as harsh family interactions with parents, academic aspiration of immigrant youths, and successful college graduation of immigrant youths), in which the drawn sample of 3344 immigrant youths is just a little bit smaller than the original wave-3 sample of CILS (N=3,613 immigrant youths). In fact, compare the sociodemographic and contextual variables to the original sample in wave 1 of CILS, no substantial differences are found between the current drawn sample (N=3344) and the original wave-1 sample (N=5262), which is now reported in the manuscript:

“As the current study was based on the sample of 3344 immigrant youths who had provided relevant information regarding the study variables across the three waves of CILS,  hence it is important to compare their differences with the original sample of 5,262 immigrant youth participants in sociodemographic and contextual variables obtained in wave-1 survey of CILS although we cannot examine their differences in the main study variables due to attrition of the original sample. Independent samples t-tests of bootstrapping by biased-corrected and accelerated (BCa) method (ĵ=1,000) showed that the drawn and original samples did not differ in age (t=1.978, p= 0.058, 95%CIs= -0.010 to 0.112), number of siblings (t=0.691, p= 0.490, 95%CIs= -0.047 to 0.975,), immigrant generations (t= -0.717, p=0.474, 95%CIs=-0.060 to 0.030,), standardized English scores (t= -1.476, p=0.130, 95%CIs= -6.705 to 0.974) and standardized math scores (t=-1.934, p= 0.520, 95%CIs= -7.672 to 0.041), and parental SES (t= -1.250, p=0.202, 95%CIs= -1.187 to 0.396,). In addition, Chi-Square tests of bootstrapping by biased-corrected and accelerated (BCa) method (ĵ=1,000) found that the drawn sample compared to the original sample had little higher proportions of female immigrant youths (54.1% vs 51.1%; X2= 6.805(1), p< 0.01, 95%CIs= 0.007 to 0.051), immigrant youths of two-parent family (70.0% vs 64.7%; X2= 23.577(1), p< 0.001, 95%CIs= 0.033 to 0.075), positive/neutral reception (92.4% vs 90.5%; X2 =8.426 (1), p< 0.01, 95%CIs= 0.011 to 0.055), and attending suburban schools (67.2% vs 63.1%; X2 =13.719(1), p< .001, 95%CIs= 0.019 to 0.064). Nevertheless, Cohence’s effects sizes showed that the significant differences are minimal, which range from w= 0.029 to 0.054. Moreover, Chi-Square tests of bootstrapping by biased-corrected and accelerated (BCa) method (ĵ=1,000) did not found differences between the drawn and original samples in ethnic origins (X2 =9.080(7), p= 0.247, 95%CIs= 0.000 to 0.034), school type ( X2 =1.910(1), p= 0.167, 95%CIs= 0.000 to 0.037), and minority-school status (X2 =0.367(1), p=0 .545, 95%CIs= 0.000 to 0.028). Due to the theoretical and empirical importance of these sociodemographic and contextual variables of immigrant youths mentioned above, immigrant youths’ gender, age, number of siblings, standardized English and math scores, family composition, parental SES, reception mode, ethnic origins, school type, school location, and minority school status were all adjusted in the modeling procedures for precluding the possibility of confounding effects.” (Lines 1241-1276 in the part of “Results”)

5) The fact that the final wave of this study was completed 20 years ago is a limitation of this study (i.e., to what extent are this study's findings still applicable in 2022?), which the authors should address in the discussion section.

Reply: Now we have reported the dataset of CILS used in the current study were collected 20 years ago as a limitation in the part of “Conclusion”, which is written:

“Third, the data of CILS were collected twenty years ago, since then the societal and economic situations of western immigrant-receiving countries have changed great differently [57, 58].” (Lines 1707-1709)

6) Future academic aspiration is solely measured by 2 items. Even though these items are quite straightforward, I wonder how much is known about the validity of these items? Would be nice if the authors could expand a bit more here.

Reply: Thank you for your a good reminder, now the operationalization and validity of using 2 relevant items to measure academic aspiration of immigrant youths have been more clearly reported:

“During adolescence youths set to ask the questions of “Who am I?” and “What will I do in the future?”, which are closely related to youths’ formation of self-concept and identity, a process referring to the establishment of the “possible self” in relation to the future [40]. Evidently, youths who have been harshly disciplined and maltreated may develop a sense of worthlessness, inferiority, and incapability [7], which may directly compromise their future hope and aspiration for academic and social success [7, 20]. Consistently, the self-system theory of motivational development posits that the general self of youths is socially constructed, especially through the process of family socialization and experiences, in which youths’ socially constructed self may act as the cognitive and motivational foundation for them to pursue future goals and success [17, 41].” (Lines 513-522 in the part of “The mediation of academic aspiration of immigrant youths”)

“Academic aspiration of immigrant youths was measured by two question items obtained in the wave-1 and 2 surveys of CILS respectively, which asked the immigrant youth 1) the highest education level he/she would like to achieve; and 2) the highest education level he/she thinks would realistically get. A 5-point scale was used to rate participants’ responses to the two items, which ranges from 1= less than high school and 5=finish a graduate degree. The two items are averaged in wave-1 and 2 surveys of CILS to represent academic aspiration of immigrant youths in early and late adolescence respectively [16, 42], in which higher scores indicate higher future academic expectations and motivations. Consistently, Di Giunta, Pastorelli, Thartori, Bombi, Baumgartner, Fabes, Martin and Enders [16] have used similar question items to tap on youths’ academic aspiration, proving its external validity. Cronbach alpha coefficients of academic aspiration of immigrant youths at wave-1 and 2 surveys of CILS were well adequate, α=.805 and .815.” (Lines 993-1004 in the part of “Measures”)

7) If I interpreted correctly, Parental hostility is based on one item? Hostility is a complex construct, I doubt whether this can be validly assessed with only one self-report item. To fully capture this complex construct, a validated questionnaire with more items on different aspects of hostility would provide much better information. I think the authors should expand more on the quality of this instrument, and reflect on this in the discussion section.

Reply: Yes, parental hostility is measured by 1 item, which is the limitation of the current study due to the parsimonious needs of conducting large longitudinal studies, like the dataset used in the current study. Nevertheless, we have mentioned its validity by referring relevant research that adopts this approach to measure parental hostility, which is written:

“Parental hostility in early adolescence was measured in wave-1 survey of CILS, in which immigrant youths were asked whether in the process of family socialization and communication with parents were full of clash and hostility from their parents. The response was based on a 4-point scale that is 1= all of the time, 2=most of the time, 3= sometimes, 4= never, which were reversely coded to indicate higher scores representing more parental hostility. Although it is better to employ validated scales to measure parental hostility, such as Parental Acceptance-Rejection Questionnaires [52], CILS is a large longitudinal survey of the life course of immigrant youths that makes it impossible to incorporate multi-item scales to measure certain single behaviors or perceptions in trading off the purpose of parsimony and avoidance of attrition [53]. This is common in large-scale longitudinal surveys. Nevertheless, recent empirical research has adopted a similar approach to tap on parental hostility and aggression in relation to adolescent delinquency and substance use [27], which appears to be methodologically valid and reliable.” (Lines 1005-1017 in the part of “Measures”)

“Moreover, as limited by the data structure of CILS, the current study only employed single or few items to measure parental hostility, emotional rejections, and harsh family interactions with parents, which is a major limitation of the current study although some existing research supports the validity of using this measurement approach to tap on minor forms of parental maltreatment [18, 27].” (Lines 1717-1765 in the part of “Conclusion”)

8) The same holds for emotional rejection, where the fact that this is a complex construct is supported by the low alpha of barely .60.

Reply: The revision has been done for parental emotional rejection now, which is written:

“Parental emotional rejection in early adolescence was measured with two items in wave-1 survey of CILS that asked immigrant youths 1) whether their parents did not like him/her much; and 2) their parents were not interested in the way he or she was, in which the items were rated by a 4-point scale ranged from 1= very true, 2=partly true, 3= not very true, and 4= not true at all. Again, although using validated scales to measure parental emotional rejection are more informative, relevant latest studies have used similar items corresponding to the current study in measuring parental emotional rejection [18, 37], supporting its reliability and validity. The correlation coefficient of the two items was r=.439, p< .001; and the Cronbach alpha was a=.611, representing an adequate level.” (Lines 1018-1028 in the part of “Measures”)   

9) Because I am not convinced that the above two measures have been operationalized in a proper way in this study, I also think that the Aggregate score 'Harsh family interactions' is of low quality. Because of the low alpha and low correlations the authors report, I would suggest not combining the above two variables into this aggregate score. The data do not support making this aggregate score in my opinion.Even though I am aware that the authors cannot change the operationalization of Emotional rejection and Parental hostility in retrospect, the authors should be much more critical on these variables and reflect on this in the discussion section.

Reply: As replied above, we now have justified the reason to combined parental hostility and emotional rejection as general harsh family interactions with parents. This is because as aforementioned they can happen separately and jointly, and based on the double- whammy thesis, if they are happened concurrently as the aggregate of harsh family interactions with parents,  it will be more harmful on educational development of immigrant youths. Hence, it is needed to examine their different and combined effects on educational development of immigrant youths according to double-whammy thesis. In fact, we have provided the following revisions to justify our using combined parental hostility and emotional rejection as aggregate of harsh family interactions with parents on educational development of immigrant youths, which include:

“Accordingly, as immigrant parents are generally of disadvantaged social status, economic deprivation, and cultural differences, they may be more prone of employing harsh discipline and unsupportive parenting [11, 27, 37]. Therefore, it is expected that experiencing parental hostility, emotional rejection, and/or harsh family interactions with parents in family is more common for immigrant youths, which is believed to detriment their educational achievement in adulthood, such as successful college graduation with a four-year undergraduate degree. As such, due to the more disadvantaged and difficult family socialization environment of immigrant youths encountered and the importance of educational success for their social mobility in adulthood as compared to their better-off local counterparts [38, 39], it is theoretically and empirically important to study how immigrant youths’ experiences of parental hostility, emotional rejection, and harsh family interactions with parents in early adolescence may adversely affect their later successful college graduation in young adulthood. In addition, immigrant youths’ experiences of parental hostility and emotional rejection may happen independently or jointly by the form of harsh family interactions with parents in family, which is congruous with the double-whammy claim that higher exposure to multiple types of harsh parental discipline and maltreatment concurrently would impair youth development further adversely [36]. Thereby, it is justifiable to examine the respective and combined effects of parental hostility, emotional rejection, and harsh family interactions with parents on educational achievement of immigrant youths in adulthood.” (Lines 203-511 in the part of “Harsh family interactions with parents and educational achievement”)

“Moreover, as limited by the data structure of CILS, the current study only employed single or few items to measure parental hostility, emotional rejections, and harsh family interactions with parents, which is a major limitation of the current study although some existing research supports the validity of using this measurement approach to tap on minor forms of parental maltreatment [18, 27].” (Lines 1699-1704 in the part of “Conclusion”)

In addition, as the construct of harsh family interactions with parents is created by combining the indicators of parental hostility and emotional rejection, in which we now use composite reliability to assess it internal consistency, which proves to be satisfactory:

“Harsh family interactions with parents in early adolescence were measured by combining the items used to indicate parental hostility and emotional rejection in aggregate. This is justified as the indicators and dimensions of parental harsh discipline and maltreatment were found closely interrelated and have more harmful effects on child and youth development when happened concurrently [6, 26, 37]. Due to harsh family interactions with parents being measured jointly by loading the indicators of parental hostility and emotional rejection together, composite reliability was used to report its internal consistency, which was pc=0.600, indicating adequate. In addition, the average correlation coefficient among the three items is well adequate, r=.316, p< .001. Hence, harsh family interactions with parents were created by combining the three items as a composite score.” (Lines 1027-1036 in the part of “Measures”)

Besides, the limitation of using just few items to measure parental hostility, emotional rejection, and harsh family interactions with parents has now reported in the part of “Conclusion”:

“Moreover, as limited by the data structure of CILS, the current study only employed single or few items to measure parental hostility, emotional rejections, and harsh family interactions with parents, which is a major limitation of the current study although some existing research supports the validity of using this measurement approach to tap on minor forms of parental maltreatment [18, 27].” (Lines 1717-1765)

10) A general note is that information on the procedure and ethical parts of the study (did an ethical review board assess the study?) could be a bit more expansive.

Reply: The current study is based on the dataset of Children of Immigrants Longitudinal Study (CILS), which is the publicly accessible data provided by the principal investigator Dr Alejandro Portes. If researchers are interested in using this dataset, they can visit The Center for Migration and Development (CMD) to download the dataset for use, which is at  https://cmd.princeton.edu/publications/data-archives/cils

For this, the current study does not need go through an official ethical review procedures of CityU where the authors belong to, but just inform the institute we use this open dataset that have been undergone ethical review at Princeton University.

11) The results section is well written, although I noted a few small issues in the way the data are presented. Table 1: Interpretation of this table would become easier if the authors report the actual percentages for the dichotomous/categorical variables (so 54.1% instead of .541 e.g.).

Reply: Thank you, and we think if we use percentages, e.g.54.1% to substitute the digit form, e.g. 0.541, which is not suitable as Table 1 also include SD and Range, hence readers can easily know what 0.541 means with reference to the Column of Range. However, if we use percentages, e.g. 54.1%, the numbers in Range will become confusing for the readers. Hence, we suggest remaining the current pattern of presentations in Table 1.

12) Information on data inspection is missing: were there missing values within this sample, and did the authors check for outliers and assumptions for the analyses? And if there were any problems, how did the authors deal with this?

Reply: Checking the assumption of normality is less tempting now as the normality assumption has been seriously quired by machaeridians nowadays.  In fact, when the sample size is large enough, like the representative sample used in the current study, the assumption of normality is not a problem, which corresponds to what Field said “We have also seen… that the central limit theorem means that as sample sizes get larger, the assumption of normality matters less because the sampling distribution will be normal regardless of what our population (or indeed sample) data look like. So, in large samples, where normality matters less (or not at all), a test of normality is more likely to be significant and make us worry about and correct for something that doesn’t need to be corrected for or worried about (p.392).” Nevertheless, when we see the variance inflation factor (VIF) values obtained in the regression modeling, which are all within the normal range, VIF=1.076 to 2.732, in which if VIF is greater than 10, the multivariate normality among the variables is a concern. For missing values, I have replied in Question 4, please refer to that reply.

Reference

Field A. (2017). Discovering statistics using IBM SPSS. London: SAGE.

13) Tables 2-4 would be easier to interpret if the authors report more details on the analysis methods that were used in the title/note of the table.

Reply: Now the titles of Table 2 to 4 have changed to “Table 2. Generalized linear mixed modeling predicting the effects of parental hostility, emotional rejection, and harsh family interactions with parents in early adolescence on immigrant youths’ college graduation in early adulthood.”, “Table 3. Generalized linear mixed modeling predicting the effects of parental hostility, emotional rejection, and harsh family interactions with parents in early adolescence on immigrant youths’ academic aspiration in late adolescence.”, and “Table 4. Generalized linear mixed modeling predicting the effects of parental hostility, emotional rejection, and harsh family interactions with parents in early adolescence and academic aspiration in late adolescence on immigrant youths’ college graduation in young adulthood.”

I hope these changes can make readers have a direct acknowledgement of what modelling procedures were use to predict the effects.

14) It is not entirely clear to me what the added value is of the separate analysis that was performed for table 2 versus table 3. I think it would make more sense to give a correlation table in table 2 and reflect on the effects of the covariates in this table, and then present table 3 for the main analysis (at least for hypotheses 1-3).

Reply: Table 2 and 3 are two not separate analyses, but they are needed to confirm the effects of parental hostility, emotional rejection, and harsh family interactions with parents, as well as academic aspiration of immigrant youths in late adolescence on successful college graduation of immigrant youths in young adulthood, in which Table 2 is used to present the effects of parental hostility, emotional rejection, and harsh family interactions with parent on successful college graduation of immigrant youths in young adulthood, in which we have not put academic aspiration of immigrant youths in late adolescence as a predictor and mediator. But table 3 is used to present the effects of parental hostility, emotional rejection, and harsh family interactions with parents, as well as academic aspiration of immigrant youths in early adolescence on academic aspiration of immigrant youths in late adolescence, which is needed to support our hypothesis 3 that academic aspiration of immigrant youths in late adolescence would mediate the effects of parental hostility, emotional rejection, and harsh family interactions with parents on immigrant youths’ college graduation in young adulthood (Table 4). Hence, of Table 2 and 3 readers can know that parental hostility, emotional rejection, and harsh family interactions with parents in early adolescence of immigrant youths can significantly predict the outcome of immigrant youths’ successful college graduation in young adulthood and their academic aspiration in late adolescence, in which academic aspiration of immigrant youths in late adolescence can be proved as a significant and important mediator, plus predictor, of the outcome of successful college graduation of immigrant youths in young adulthood, which can be viewed in Table 4.

For giving a correlation table as Table 2, I think it would make the whole study interminable and unnecessary. This is because if the important predictors of parental hostility, emotional rejection, and harsh family interactions with parents, as well as academic aspiration of immigrant youths in late adolescence can significantly predict the outcome of successful graduation of immigrant youths in young adulthood by generalized linear mixed modelling procedures, which mean that their significant correlations must be confirmed in correlation analysis if we simply put them together in a bivariate association by correlation analysis.

15) The sentence in lines 444-446 is a bit overstated in my opinion. Not ‘any forms of...’ is assessed in this study, only emotional forms of harsh parenting.

Reply: Now the sentence of “However, this study found that any forms of parental maltreatment and parental harshness occasioned long-lasting detriments to positive development of immigrant youths across different ethnic origins”  has changed to “However, this study found that parental hostility, emotional rejection, and harsh family interactions with parents experienced by immigrant youths in early adolescence occasioned long-lasting detriments to positive educational development of immigrant youths in terms of development of academic aspiration in late adolescence and educational success in young adulthood across different ethnic origins.” I hope this change can make the presentation of the findings more accurate and precise.

16) The statement in lines 449-452 is not completely correct: emotional neglect is the most prevalent form of child maltreatment, it would be good if the authors would refer to recent large prevalence studies for this statement.

Reply: Now the whole manuscript has been thoroughly revised to let readers know we focus on investigating the effects of parental hostility, emotional rejection, and harsh family interactions with parents, as well as academic aspiration of immigrant youths in late adolescence on successful college graduation of immigrant youth in young adulthood, in which we have stated that this study aims to examine these minor forms of harsh family socialization experiences on adverse educational development of immigrant youths.

17) Hostility, rejection and harsh family interactions are presented as three different variables, which is not consistent with how it was tested in this study (see also my comments in the method section: I do not think it is justified to speak of 'harsh family interactions' as aggregate variable).

Reply: Parental hostility and emotional rejection are investigated as two different types of harsh family socialization experiences of immigrant youths, in which they can be independently or jointly exist in the life of immigrant youths. Hence in this study we want to examine their combined effect as the aggregate of harsh family interactions with parents on educational development of immigrant youths, in which according to the double-whammy thesis we expect harsh family interactions with parents would have the strongest effects on educational development of immigrant youths, which is apparently supported in the current study. In fact, the whole manuscript has now been thoroughly rewritten to make the respective investigations of parental hostility, emotional rejection, and harsh family interactions with parents in contribution to educational development of immigrant youths more justifiable.

18) The limitations of this study are discussed too briefly, and the authors should be more reflective on several important limitations which are currently not mentioned: the operationalization of the 'harsh family interactions' variables and the fact that data collection took place over 20 years ago.

Reply: Now the limitations of the study has been strengthened, which are written as

“Nevertheless, several limitations exist in this study. First, immigrant youths were recruited mainly from the two immigrant-receiving regions of the United States although CILS contains a large and representative sample. Second, this study only examined the effects of parental hostility, emotional rejection, and harsh family interactions with parents in early adolescence of immigrant youths on successful college graduation of immigrant youths in young adulthood through the mediation of their development of academic aspiration in late adolescence. However, other serious types of parental maltreatment, e.g. sexual and physical abuse, and some important intrapersonal constructs of immigrant youths, such as self-esteem and life meaning that may mediate the study relationships, have not examined. Thereby, future research should investigate accumulative effects of different maltreatment experiences of immigrant youths in relation to their educational achievement through the development of multiple cognitive and intrapersonal mediators among immigrant youths. Third, the data of CILS were collected twenty years ago, since then the societal and economic situations of western immigrant-receiving countries have changed great differently [57, 58]. Therefore, more updated data of immigration research are needed to further confirm the relationships between harsh family socialization experiences of immigrant youths and their educational development nowadays. Moreover, as limited by the data structure of CILS, the current study only employed single or few items to measure parental hostility, emotional rejections, and harsh family interactions with parents, which is a major limitation of the current study although some existing research supports the validity of using this measurement approach to tap on minor forms of parental maltreatment [18, 27]. Lastly but not least, as the data of CILS did not include non-Hispanic White youths who distinctively differ to immigrant youths culturally, economically, and socially, cross-population validity and comparison are hence impossible in this study. In all, it is believed that if timely and adequate interventions and service supports are provided to harshly disciplined and maltreated immigrant youths and their families, these youths can also be thriving academically and socially.” (Lines 1703-1779 in the part of “Conclusion”)

19) The authors did a good job explaining the relevance of their study and translating the study's results into implications for society and future studies.

Reply: Thank you again.

Reviewer 2 Report

Summary

This paper investigated whether the effects of harsh parenting on academic achievements of immigrant youth is mediated by their academic aspirations. The study used a longitudinal design, following young adolescents into adulthood. The longitudinal design of the study and the large sample size are important strengths. In addition, the hypothesized mediation effects are theoretically sound and can lead to relevant implications for society as well as future studies. The manuscript is generally well-written and the line of thought is easy to follow. However, my most important concern is the operationalization of parental hostility measures, which make me much more cautious than the authors to draw firm conclusions from this report. My concerns are outlined below, summarized per section of the manuscript.

Introduction

Generally well-written introduction which makes the relevance and unique contribution of the proposed study clear. The relevant theories are well-introduced and the line of thought is easy to follow. I only have a few minor feedback points for the introduction:

For the hypotheses, I wonder what the added value is of looking at the two 'subscales' of the 'harsh family interactions' variable separately, if the final hypotheses concern the composite of these two.

The variables mentioned in lines 147-150 (previous paragraph) are not mentioned as covariates under Present study, even though the analyses do include these variables as covariates.

Methods

The original sample comprised of 5262 participants, this report included only 3344. Can the authors expand a bit more about the representativeness of this sample? E.g., is there any information on differences/reasons why other participants dropped out and how might this affect this study’s results?

The fact that the final wave of this study was completed 20 years ago is a limitation of this study (i.e., to what extent are this study's findings still applicable in 2022?), which the authors should address in the discussion section.

Future academic aspiration is solely measured by 2 items. Even though these items are quite straightforward, I wonder how much is known about the validity of these items? Would be nice if the authors could expand a bit more here.

If I interpreted correctly, Parental hostility is based on one item? Hostility is a complex construct, I doubt whether this can be validly assessed with only one self-report item. To fully capture this complex construct, a validated questionnaire with more items on different aspects of hostility would provide much better information. I think the authors should expand more on the quality of this instrument, and reflect on this in the discussion section.

The same holds for emotional rejection, where the fact that this is a complex construct is supported by the low alpha of barely .60.

Because I am not convinced that the above two measures have been operationalized in a proper way in this study, I also think that the Aggregate score 'Harsh family interactions' is of low quality. Because of the low alpha and low correlations the authors report, I would suggest not combining the above two variables into this aggregate score. The data do not support making this aggregate score in my opinion.

Even though I am aware that the authors cannot change the operationalization of Emotional rejection and Parental hostility in retrospect, the authors should be much more critical on these variables and reflect on this in the discussion section.

A general note is that information on the procedure and ethical parts of the study (did an ethical review board assess the study?) could be a bit more expansive.

Results

The results section is well written, although I noted a few small issues in the way the data are presented.

Table 1: Interpretation of this table would become easier if the authors report the actual percentages for the dichotomous/categorical variables (so 54.1% instead of .541 e.g.).

Information on data inspection is missing: were there missing values within this sample, and did the authors check for outliers and assumptions for the analyses? And if there were any problems, how did the authors deal with this?

Tables 2-4 would be easier to interpret if the authors report more details on the analysis methods that were used in the title/note of the table.

It is not entirely clear to me what the added value is of the separate analysis that was performed for table 2 versus table 3. I think it would make more sense to give a correlation table in table 2 and reflect on the effects of the covariates in this table, and then present table 3 for the main analysis (at least for hypotheses 1-3).

Discussion

The sentence in lines 444-446 is a bit overstated in my opinion. Not ‘any forms of...’ is assessed in this study, only emotional forms of harsh parenting.

The statement in lines 449-452 is not completely correct: emotional neglect is the most prevalent form of child maltreatment, it would be good if the authors would refer to recent large prevalence studies for this statement.

Hostility, rejection and harsh family interactions are presented as three different variables, which is not consistent with how it was tested in this study (see also my comments in the method section: I do not think it is justified to speak of 'harsh family interactions' as aggregate variable).

The limitations of this study are discussed too briefly, and the authors should be more reflective on several important limitations which are currently not mentioned: the operationalization of the 'harsh family interactions' variables and the fact that data collection took place over 20 years ago.

The authors did a good job explaining the relevance of their study and translating the study's results into implications for society and future studies.

Author Response

Thank for reviewing our submitted manuscript titled “Harsh Family Interactions with Parents and Educational Achievement of Immigrant Youths in young adulthood: A Longitudinal Study”. We have revised the whole manuscript according to the comments of the reviewers and our responses are follows:

For reviewer 2

1) The manuscript is generally well-written and the line of thought is easy to follow. However, my most important concern is the operationalization of parental hostility measures, which make me much more cautious than the authors to draw firm conclusions from this report.  

Reply: Thank you, for the operationalization of parental hostility, emotional rejection, and their aggregate as harsh family interactions with parents, which are now thoroughly revised and clearly presented to justify its meanings, reasons, influences and limitations by theories, past empirical research, and rational considerations based on the available information provided by the Children of Immigrants Longitudinal Study (CILS), where the current study obtain its dataset. For details of the revisions, please refer to the parts of “Harsh family interactions with parents and educational achievement” and “The mediation of academic aspiration of immigrant youths”, and “The present study”, as well as “Conclusion (Lines: 1709-1717)”. In fact, we now also have rewritten the presentations of the effects of parental hostility, emotional rejection, and their aggregate as harsh family interactions with parents on educational development of immigrant youths in the parts of “Discussion” and “Conclusion” to make the contents and presentations more accurate and valid, which include:

“This study is the first attempt to investigate how harsh family experiences of immigrant youths in early adolescence adversely impact their educational achievement in young adulthood through the mediation of academic aspiration of immigrant youths in late adolescence, which is of research importance due to disadvantaged social conditions and cultural differences of immigrant families [13, 49]. Some scholars reckoned that harsh parental discipline and strict parenting are common and even regarded as “normative” in immigrant families, which are thought less harmful to development of immigrant youths [9, 11]. However, this study found that parental hostility, emotional rejection, and harsh family interactions with parents experienced by immigrant youths in early adolescence occasioned long-lasting detriments to positive educational development of immigrant youths in terms of development of academic aspiration in late adolescence and educational success in young adulthood across different ethnic origins. It is noted that parental hostility and emotional rejection, or their aggregate as harsh family interactions with parents, are common in many disadvantaged families, especially those of immigrant background [5, 11]. This accords with some recent research suggesting that parental hostility and emotional indifference and rejection are common forms of parental maltreatment experienced by children and youths in disadvantaged and immigrant families [6, 9, 10, 19]. In fact, as minor forms of parental maltreatment are hard to define and identify, which makes it easily overlooked by policy makers, teachers, human service practitioners, and researchers. Nevertheless, limited relevant research recently found that these minor forms of parental maltreatment are far less from negligibility than those severe parental maltreatment types due to their frequency, chronicity, and incessancy [6, 12]. It is supported in this study that parental hostility, emotional rejection, and harsh family interactions with parents in early adolescence significantly impaired the likelihood of immigrant youths’ successful college graduation in young adulthood, bearing profound implications for their later life transformation and social mobility [10, 32].” (Lines 1501-1571 in the part of “Discussion”)

“Moreover, as limited by the data structure of CILS, the current study only employed single or few items to measure parental hostility, emotional rejections, and harsh family interactions with parents, which is a major limitation of the current study although some existing research supports the validity of using this measurement approach to tap on minor forms of parental maltreatment [18, 27].” (Lines 1711-1759 in the part of “Conclusion”)

2) For the hypotheses, I wonder what the added value is of looking at the two 'subscales' of the 'harsh family interactions' variable separately, if the final hypotheses concern the composite of these two.

Reply: The current study investigates how parental hostility, emotional rejection, and their aggregate as harsh family interactions with parents may negatively affect successful college graduation of immigrant youths separately and jointly is necessary. This is because they can happen independently and/or concurrently. Hence, compare and investigate the independent and joint effects of parental hostility and emotional rejection on educational development of immigrant youths are academically and socially important, which accords with double-whammy thesis that is to anticipate higher exposure to multiple types of harsh parental discipline and maltreatment would impair youth development more serious although different types of  multiple types of harsh parental discipline and maltreatment may sometimes happen separately. For this, we now have rewritten the relevant presentations to:

“Nevertheless, parental hostility and emotional rejection or aggregated as a general form of harsh family interactions with parents in family have not received adequate research attention, which are mainly due to their differences to physical and sexual abuse and are regarded as a minor form of parental maltreatment or just harsh discipline [23, 24]. This corresponds to what Allan, et al. [25] mentioned “parental hostility, as opposed to physical or sexual abuse and neglect, has not received much attention. …Abuse generally consists of physical or sexual brutality whereas hostility generally consists of verbal and psychological abusiveness such as anger or antagonism (p.169).” In addition, Khaleque [26] stated that “(p)arental (emotional) rejection, on the other hand, refers to the withdrawal or lack of parental warmth, affection, care, comfort, nurturance, support, or love toward their children (p.977).” Hence, it is justified to conceptualize parental hostility in this study as the manifestation of conflicting and clashing relationships from parents toward their youth children [25, 27], and parental emotional rejection as parental display of absent interest, warmth, love, and nurturance toward their youth children in an aversive and unapproved way [28, 29], and harsh family interactions with parents as the aggregate of these conflicting, clashing, rejecting, apathetic, and disapproving processes of parent-youth interactions [25, 28, 30].” (Lines 92-108 in the part of “Harsh family interactions with parents and educational achievement”)

“Yet, although limited pertinent research has reported the harmful impacts of parental hostility and rejection on behavioral and psychological adjustments of children and youths, little is known about how these negative family socialization experiences may negatively affect educational development of immigrant youths. This is research-worthy from the model of family socialization that youths experiencing harsh discipline and maltreatment implies their receiving low parental responsiveness and support and high parental negativity [22, 33, 34], which are harmful to youth health development educationally and socially.” (Lines 175-182 in the part of “Harsh family interactions with parents and educational achievement”)

“Accordingly, as immigrant parents are generally of disadvantaged social status, economic deprivation, and cultural differences, they may be more prone of employing harsh discipline and unsupportive parenting [11, 27, 37]. Therefore, it is expected that experiencing parental hostility, emotional rejection, and/or harsh family interactions with parents in family is more common for immigrant youths, which is believed to detriment their educational achievement in adulthood, such as successful college graduation with a four-year undergraduate degree. As such, due to the more disadvantaged and difficult family socialization environment of immigrant youths encountered and the importance of educational success for their social mobility in adulthood as compared to their better-off local counterparts [38, 39], it is theoretically and empirically important to study how immigrant youths’ experiences of parental hostility, emotional rejection, and harsh family interactions with parents in early adolescence may adversely affect their later successful college graduation in young adulthood. In addition, immigrant youths’ experiences of parental hostility and emotional rejection may happen independently or jointly by the form of harsh family interactions with parents in family, which is congruous with the double-whammy claim that higher exposure to multiple types of harsh parental discipline and maltreatment concurrently would impair youth development further adversely [36]. Thereby, it is justifiable to examine the respective and combined effects of parental hostility, emotional rejection, and harsh family interactions with parents on educational achievement of immigrant youths in adulthood.” (Lines 203-511 in the part of “Harsh family interactions with parents and educational achievement”)

“Moreover, as limited by the data structure of CILS, the current study only employed single or few items to measure parental hostility, emotional rejections, and harsh family interactions with parents, which is a major limitation of the current study although some existing research supports the validity of using this measurement approach to tap on minor forms of parental maltreatment [18, 27].” (Lines 1699-1704 in the part of “Conclusion”)

Besides, the whole manuscript has been thoroughly revised to make the conceptualization and operationalization of parental hostility, emotional rejection, and harsh family interactions with parents more justifiable and understandable, especially in the part of “Measures (Lines 981-1030)”.

In sum, we would like to confirm in the current study that the minor forms of parental hostility and emotional rejection would adversely impact educational development of immigrant youths longitudinally, and their aggregate as harsh family interactions with parents would be more harmful on educational development of immigrant youths. Accordingly, our results generally support this double-whammy postulation.

3) The variables mentioned in lines 147-150 (previous paragraph) are not mentioned as covariates under Present study, even though the analyses do include these variables as covariates.

Reply: The variables have been stated as covariates, in which their impacts on educational motivation and success of immigrant youths have further elaborated more clearly now:

“Hence, in this study, we incorporate the contextual effects of family composition, parental SES, and mode of reception at individual level and school type, school minority status, and school location at school level as covariates when examining the study relationships of parental hostility, emotional rejection, and harsh family interactions with parents in contribution to academic aspiration of immigrant youths and their successful college graduation in young adulthood.” (Lines 808-812 in the part of “Contextual influences and educational achievement of immigrant youths”)

“Moreover, this study adjusted for gender, age, number of siblings, generation status, standardized English and math test scores and ethnic origins of immigrant youths for precluding sociodemographic confounding effects. Immigration research reported that female immigrant youths generally performed better academically than their male counterparts [47], and older aged immigrant youths had more difficulties in school adjustment than their younger counterparts [32]. Besides, immigrant youths of more siblings connote their family supports and care are diluted, which may compromise their school performance [32, 42]. For immigrant generation status, earlier generations of immigrant youths are reported to have higher educational motivation and performance than their later generations, albeit their economic and social disadvantages, which is called as “generation mystery” [48]. Additionally, ethnic origins of immigrant youths are found to have different effects on their educational achievement, in which immigrant youths of Northeast/ Eastern Asian origin showed higher educational success and Mexican and Caribbean immigrant youths manifestly fell behind academically while other ethnic groups were in-between [42, 49]. Therefore, immigrant youths in this study are classified into eight major ethnic groups with reference to their cultural and geographic adjacency [32, 49], which include Mexican, Caribbean, Cuban, Central and Southern American, Southeast Asian, Northeast/ Eastern Asian, Middle Eastern and African, European origins.” (Lines 919-936 in the part of “The present study”)

And their effects on academic aspiration of immigrant youths and their successful college graduation are also more clearly reported:

“For the effects of sociodemographic and contextual covariates on successful college graduation of immigrant youths in young adulthood, female immigrant youths, as compared to their male counterparts, had significantly the higher odds of successful college graduation across the three mixed-effects models by 23.9% to 25.7%, and older aged immigrant youths generally showed the reduced odds of completing college education by 7.4% to 7.9%. Moreover, 1.5- and 2-generation immigrant youths, when comparing to their 2.5-gneration peers, unanimously had the greater odds of college graduation by 46.1% to 61.4%. More important, immigrant youths from intact family and of better parental SES were more likely to obtain a college degree than their counterparts from broken and economically poorer family by the higher odds of 15.9% to 17.4% for the former and around 51% for the latter. For school-level variables, immigrant youths attending public secondary schools significantly had the lower odds of successful college gradation by around 13% as compared to their counterparts from private schools. Furthermore, as compared to their counterparts from rural and suburban schools, immigrant youths studying in inner-city schools significantly had the lower odds of college graduation by 35.1% to 37%. Lastly, ethnic origins did not show significant effects on educational success of immigrant youths in young adulthood, except Middle-East and African immigrant youths, as compared to their Northeast/ Eastern Asian counterparts, had the greater odds of completing college education by 2.145 to 2.307 times.” (Lines 1316-1334 in the part of “Results”).

“Examining the effects of sociodemographic and contextual covariates, female and 1.5- and 2-generation immigrant youths significantly had higher academic aspiration in late adolescence than their male and 2.5-gernation counterparts. Besides, immigrant youths from intact family and of better parental SES significantly had higher academic aspiration in late adolescence than their counterparts from broken and economically poorer family. What’s more, public school and inner-city school statuses significantly contributed to lower academic aspiration of immigrant youths in late adolescence at school level.” (Lines 1347-1370 in the part of “Results”).

4) The original sample comprised of 5262 participants, this report included only 3344. Can the authors expand a bit more about the representativeness of this sample? E.g., is there any information on differences/reasons why other participants dropped out and how might this affect this study’s results?

Reply: The current study employs the sample of 3344 immigrant youths who had provided valid information regarding the study variables (parental hostility, emotional rejection, and their aggregate as harsh family interactions with parents, academic aspiration of immigrant youths, and successful college graduation of immigrant youths), in which the drawn sample of 3344 immigrant youths is just a little bit smaller than the original wave-3 sample of CILS (N=3,613 immigrant youths). In fact, compare the sociodemographic and contextual variables to the original sample in wave 1 of CILS, no substantial differences are found between the current drawn sample (N=3344) and the original wave-1 sample (N=5262), which is now reported in the manuscript:

“As the current study was based on the sample of 3344 immigrant youths who had provided relevant information regarding the study variables across the three waves of CILS,  hence it is important to compare their differences with the original sample of 5,262 immigrant youth participants in sociodemographic and contextual variables obtained in wave-1 survey of CILS although we cannot examine their differences in the main study variables due to attrition of the original sample. Independent samples t-tests of bootstrapping by biased-corrected and accelerated (BCa) method (ĵ=1,000) showed that the drawn and original samples did not differ in age (t=1.978, p= 0.058, 95%CIs= -0.010 to 0.112), number of siblings (t=0.691, p= 0.490, 95%CIs= -0.047 to 0.975,), immigrant generations (t= -0.717, p=0.474, 95%CIs=-0.060 to 0.030,), standardized English scores (t= -1.476, p=0.130, 95%CIs= -6.705 to 0.974) and standardized math scores (t=-1.934, p= 0.520, 95%CIs= -7.672 to 0.041), and parental SES (t= -1.250, p=0.202, 95%CIs= -1.187 to 0.396,). In addition, Chi-Square tests of bootstrapping by biased-corrected and accelerated (BCa) method (ĵ=1,000) found that the drawn sample compared to the original sample had little higher proportions of female immigrant youths (54.1% vs 51.1%; X2= 6.805(1), p< 0.01, 95%CIs= 0.007 to 0.051), immigrant youths of two-parent family (70.0% vs 64.7%; X2= 23.577(1), p< 0.001, 95%CIs= 0.033 to 0.075), positive/neutral reception (92.4% vs 90.5%; X2 =8.426 (1), p< 0.01, 95%CIs= 0.011 to 0.055), and attending suburban schools (67.2% vs 63.1%; X2 =13.719(1), p< .001, 95%CIs= 0.019 to 0.064). Nevertheless, Cohence’s effects sizes showed that the significant differences are minimal, which range from w= 0.029 to 0.054. Moreover, Chi-Square tests of bootstrapping by biased-corrected and accelerated (BCa) method (ĵ=1,000) did not found differences between the drawn and original samples in ethnic origins (X2 =9.080(7), p= 0.247, 95%CIs= 0.000 to 0.034), school type ( X2 =1.910(1), p= 0.167, 95%CIs= 0.000 to 0.037), and minority-school status (X2 =0.367(1), p=0 .545, 95%CIs= 0.000 to 0.028). Due to the theoretical and empirical importance of these sociodemographic and contextual variables of immigrant youths mentioned above, immigrant youths’ gender, age, number of siblings, standardized English and math scores, family composition, parental SES, reception mode, ethnic origins, school type, school location, and minority school status were all adjusted in the modeling procedures for precluding the possibility of confounding effects.” (Lines 1241-1276 in the part of “Results”)

5) The fact that the final wave of this study was completed 20 years ago is a limitation of this study (i.e., to what extent are this study's findings still applicable in 2022?), which the authors should address in the discussion section.

Reply: Now we have reported the dataset of CILS used in the current study were collected 20 years ago as a limitation in the part of “Conclusion”, which is written:

“Third, the data of CILS were collected twenty years ago, since then the societal and economic situations of western immigrant-receiving countries have changed great differently [57, 58].” (Lines 1707-1709)

6) Future academic aspiration is solely measured by 2 items. Even though these items are quite straightforward, I wonder how much is known about the validity of these items? Would be nice if the authors could expand a bit more here.

Reply: Thank you for your a good reminder, now the operationalization and validity of using 2 relevant items to measure academic aspiration of immigrant youths have been more clearly reported:

“During adolescence youths set to ask the questions of “Who am I?” and “What will I do in the future?”, which are closely related to youths’ formation of self-concept and identity, a process referring to the establishment of the “possible self” in relation to the future [40]. Evidently, youths who have been harshly disciplined and maltreated may develop a sense of worthlessness, inferiority, and incapability [7], which may directly compromise their future hope and aspiration for academic and social success [7, 20]. Consistently, the self-system theory of motivational development posits that the general self of youths is socially constructed, especially through the process of family socialization and experiences, in which youths’ socially constructed self may act as the cognitive and motivational foundation for them to pursue future goals and success [17, 41].” (Lines 513-522 in the part of “The mediation of academic aspiration of immigrant youths”)

“Academic aspiration of immigrant youths was measured by two question items obtained in the wave-1 and 2 surveys of CILS respectively, which asked the immigrant youth 1) the highest education level he/she would like to achieve; and 2) the highest education level he/she thinks would realistically get. A 5-point scale was used to rate participants’ responses to the two items, which ranges from 1= less than high school and 5=finish a graduate degree. The two items are averaged in wave-1 and 2 surveys of CILS to represent academic aspiration of immigrant youths in early and late adolescence respectively [16, 42], in which higher scores indicate higher future academic expectations and motivations. Consistently, Di Giunta, Pastorelli, Thartori, Bombi, Baumgartner, Fabes, Martin and Enders [16] have used similar question items to tap on youths’ academic aspiration, proving its external validity. Cronbach alpha coefficients of academic aspiration of immigrant youths at wave-1 and 2 surveys of CILS were well adequate, α=.805 and .815.” (Lines 993-1004 in the part of “Measures”)

7) If I interpreted correctly, Parental hostility is based on one item? Hostility is a complex construct, I doubt whether this can be validly assessed with only one self-report item. To fully capture this complex construct, a validated questionnaire with more items on different aspects of hostility would provide much better information. I think the authors should expand more on the quality of this instrument, and reflect on this in the discussion section.

Reply: Yes, parental hostility is measured by 1 item, which is the limitation of the current study due to the parsimonious needs of conducting large longitudinal studies, like the dataset used in the current study. Nevertheless, we have mentioned its validity by referring relevant research that adopts this approach to measure parental hostility, which is written:

“Parental hostility in early adolescence was measured in wave-1 survey of CILS, in which immigrant youths were asked whether in the process of family socialization and communication with parents were full of clash and hostility from their parents. The response was based on a 4-point scale that is 1= all of the time, 2=most of the time, 3= sometimes, 4= never, which were reversely coded to indicate higher scores representing more parental hostility. Although it is better to employ validated scales to measure parental hostility, such as Parental Acceptance-Rejection Questionnaires [52], CILS is a large longitudinal survey of the life course of immigrant youths that makes it impossible to incorporate multi-item scales to measure certain single behaviors or perceptions in trading off the purpose of parsimony and avoidance of attrition [53]. This is common in large-scale longitudinal surveys. Nevertheless, recent empirical research has adopted a similar approach to tap on parental hostility and aggression in relation to adolescent delinquency and substance use [27], which appears to be methodologically valid and reliable.” (Lines 1005-1017 in the part of “Measures”)

“Moreover, as limited by the data structure of CILS, the current study only employed single or few items to measure parental hostility, emotional rejections, and harsh family interactions with parents, which is a major limitation of the current study although some existing research supports the validity of using this measurement approach to tap on minor forms of parental maltreatment [18, 27].” (Lines 1717-1765 in the part of “Conclusion”)

8) The same holds for emotional rejection, where the fact that this is a complex construct is supported by the low alpha of barely .60.

Reply: The revision has been done for parental emotional rejection now, which is written:

“Parental emotional rejection in early adolescence was measured with two items in wave-1 survey of CILS that asked immigrant youths 1) whether their parents did not like him/her much; and 2) their parents were not interested in the way he or she was, in which the items were rated by a 4-point scale ranged from 1= very true, 2=partly true, 3= not very true, and 4= not true at all. Again, although using validated scales to measure parental emotional rejection are more informative, relevant latest studies have used similar items corresponding to the current study in measuring parental emotional rejection [18, 37], supporting its reliability and validity. The correlation coefficient of the two items was r=.439, p< .001; and the Cronbach alpha was a=.611, representing an adequate level.” (Lines 1018-1028 in the part of “Measures”)   

9) Because I am not convinced that the above two measures have been operationalized in a proper way in this study, I also think that the Aggregate score 'Harsh family interactions' is of low quality. Because of the low alpha and low correlations the authors report, I would suggest not combining the above two variables into this aggregate score. The data do not support making this aggregate score in my opinion.Even though I am aware that the authors cannot change the operationalization of Emotional rejection and Parental hostility in retrospect, the authors should be much more critical on these variables and reflect on this in the discussion section.

Reply: As replied above, we now have justified the reason to combined parental hostility and emotional rejection as general harsh family interactions with parents. This is because as aforementioned they can happen separately and jointly, and based on the double- whammy thesis, if they are happened concurrently as the aggregate of harsh family interactions with parents,  it will be more harmful on educational development of immigrant youths. Hence, it is needed to examine their different and combined effects on educational development of immigrant youths according to double-whammy thesis. In fact, we have provided the following revisions to justify our using combined parental hostility and emotional rejection as aggregate of harsh family interactions with parents on educational development of immigrant youths, which include:

“Accordingly, as immigrant parents are generally of disadvantaged social status, economic deprivation, and cultural differences, they may be more prone of employing harsh discipline and unsupportive parenting [11, 27, 37]. Therefore, it is expected that experiencing parental hostility, emotional rejection, and/or harsh family interactions with parents in family is more common for immigrant youths, which is believed to detriment their educational achievement in adulthood, such as successful college graduation with a four-year undergraduate degree. As such, due to the more disadvantaged and difficult family socialization environment of immigrant youths encountered and the importance of educational success for their social mobility in adulthood as compared to their better-off local counterparts [38, 39], it is theoretically and empirically important to study how immigrant youths’ experiences of parental hostility, emotional rejection, and harsh family interactions with parents in early adolescence may adversely affect their later successful college graduation in young adulthood. In addition, immigrant youths’ experiences of parental hostility and emotional rejection may happen independently or jointly by the form of harsh family interactions with parents in family, which is congruous with the double-whammy claim that higher exposure to multiple types of harsh parental discipline and maltreatment concurrently would impair youth development further adversely [36]. Thereby, it is justifiable to examine the respective and combined effects of parental hostility, emotional rejection, and harsh family interactions with parents on educational achievement of immigrant youths in adulthood.” (Lines 203-511 in the part of “Harsh family interactions with parents and educational achievement”)

“Moreover, as limited by the data structure of CILS, the current study only employed single or few items to measure parental hostility, emotional rejections, and harsh family interactions with parents, which is a major limitation of the current study although some existing research supports the validity of using this measurement approach to tap on minor forms of parental maltreatment [18, 27].” (Lines 1699-1704 in the part of “Conclusion”)

In addition, as the construct of harsh family interactions with parents is created by combining the indicators of parental hostility and emotional rejection, in which we now use composite reliability to assess it internal consistency, which proves to be satisfactory:

“Harsh family interactions with parents in early adolescence were measured by combining the items used to indicate parental hostility and emotional rejection in aggregate. This is justified as the indicators and dimensions of parental harsh discipline and maltreatment were found closely interrelated and have more harmful effects on child and youth development when happened concurrently [6, 26, 37]. Due to harsh family interactions with parents being measured jointly by loading the indicators of parental hostility and emotional rejection together, composite reliability was used to report its internal consistency, which was pc=0.600, indicating adequate. In addition, the average correlation coefficient among the three items is well adequate, r=.316, p< .001. Hence, harsh family interactions with parents were created by combining the three items as a composite score.” (Lines 1027-1036 in the part of “Measures”)

Besides, the limitation of using just few items to measure parental hostility, emotional rejection, and harsh family interactions with parents has now reported in the part of “Conclusion”:

“Moreover, as limited by the data structure of CILS, the current study only employed single or few items to measure parental hostility, emotional rejections, and harsh family interactions with parents, which is a major limitation of the current study although some existing research supports the validity of using this measurement approach to tap on minor forms of parental maltreatment [18, 27].” (Lines 1717-1765)

10) A general note is that information on the procedure and ethical parts of the study (did an ethical review board assess the study?) could be a bit more expansive.

Reply: The current study is based on the dataset of Children of Immigrants Longitudinal Study (CILS), which is the publicly accessible data provided by the principal investigator Dr Alejandro Portes. If researchers are interested in using this dataset, they can visit The Center for Migration and Development (CMD) to download the dataset for use, which is at  https://cmd.princeton.edu/publications/data-archives/cils

For this, the current study does not need go through an official ethical review procedures of CityU where the authors belong to, but just inform the institute we use this open dataset that have been undergone ethical review at Princeton University.

11) The results section is well written, although I noted a few small issues in the way the data are presented. Table 1: Interpretation of this table would become easier if the authors report the actual percentages for the dichotomous/categorical variables (so 54.1% instead of .541 e.g.).

Reply: Thank you, and we think if we use percentages, e.g.54.1% to substitute the digit form, e.g. 0.541, which is not suitable as Table 1 also include SD and Range, hence readers can easily know what 0.541 means with reference to the Column of Range. However, if we use percentages, e.g. 54.1%, the numbers in Range will become confusing for the readers. Hence, we suggest remaining the current pattern of presentations in Table 1.

12) Information on data inspection is missing: were there missing values within this sample, and did the authors check for outliers and assumptions for the analyses? And if there were any problems, how did the authors deal with this?

Reply: Checking the assumption of normality is less tempting now as the normality assumption has been seriously quired by machaeridians nowadays.  In fact, when the sample size is large enough, like the representative sample used in the current study, the assumption of normality is not a problem, which corresponds to what Field said “We have also seen… that the central limit theorem means that as sample sizes get larger, the assumption of normality matters less because the sampling distribution will be normal regardless of what our population (or indeed sample) data look like. So, in large samples, where normality matters less (or not at all), a test of normality is more likely to be significant and make us worry about and correct for something that doesn’t need to be corrected for or worried about (p.392).” Nevertheless, when we see the variance inflation factor (VIF) values obtained in the regression modeling, which are all within the normal range, VIF=1.076 to 2.732, in which if VIF is greater than 10, the multivariate normality among the variables is a concern. For missing values, I have replied in Question 4, please refer to that reply.

Reference

Field A. (2017). Discovering statistics using IBM SPSS. London: SAGE.

13) Tables 2-4 would be easier to interpret if the authors report more details on the analysis methods that were used in the title/note of the table.

Reply: Now the titles of Table 2 to 4 have changed to “Table 2. Generalized linear mixed modeling predicting the effects of parental hostility, emotional rejection, and harsh family interactions with parents in early adolescence on immigrant youths’ college graduation in early adulthood.”, “Table 3. Generalized linear mixed modeling predicting the effects of parental hostility, emotional rejection, and harsh family interactions with parents in early adolescence on immigrant youths’ academic aspiration in late adolescence.”, and “Table 4. Generalized linear mixed modeling predicting the effects of parental hostility, emotional rejection, and harsh family interactions with parents in early adolescence and academic aspiration in late adolescence on immigrant youths’ college graduation in young adulthood.”

I hope these changes can make readers have a direct acknowledgement of what modelling procedures were use to predict the effects.

14) It is not entirely clear to me what the added value is of the separate analysis that was performed for table 2 versus table 3. I think it would make more sense to give a correlation table in table 2 and reflect on the effects of the covariates in this table, and then present table 3 for the main analysis (at least for hypotheses 1-3).

Reply: Table 2 and 3 are two not separate analyses, but they are needed to confirm the effects of parental hostility, emotional rejection, and harsh family interactions with parents, as well as academic aspiration of immigrant youths in late adolescence on successful college graduation of immigrant youths in young adulthood, in which Table 2 is used to present the effects of parental hostility, emotional rejection, and harsh family interactions with parent on successful college graduation of immigrant youths in young adulthood, in which we have not put academic aspiration of immigrant youths in late adolescence as a predictor and mediator. But table 3 is used to present the effects of parental hostility, emotional rejection, and harsh family interactions with parents, as well as academic aspiration of immigrant youths in early adolescence on academic aspiration of immigrant youths in late adolescence, which is needed to support our hypothesis 3 that academic aspiration of immigrant youths in late adolescence would mediate the effects of parental hostility, emotional rejection, and harsh family interactions with parents on immigrant youths’ college graduation in young adulthood (Table 4). Hence, of Table 2 and 3 readers can know that parental hostility, emotional rejection, and harsh family interactions with parents in early adolescence of immigrant youths can significantly predict the outcome of immigrant youths’ successful college graduation in young adulthood and their academic aspiration in late adolescence, in which academic aspiration of immigrant youths in late adolescence can be proved as a significant and important mediator, plus predictor, of the outcome of successful college graduation of immigrant youths in young adulthood, which can be viewed in Table 4.

For giving a correlation table as Table 2, I think it would make the whole study interminable and unnecessary. This is because if the important predictors of parental hostility, emotional rejection, and harsh family interactions with parents, as well as academic aspiration of immigrant youths in late adolescence can significantly predict the outcome of successful graduation of immigrant youths in young adulthood by generalized linear mixed modelling procedures, which mean that their significant correlations must be confirmed in correlation analysis if we simply put them together in a bivariate association by correlation analysis.

15) The sentence in lines 444-446 is a bit overstated in my opinion. Not ‘any forms of...’ is assessed in this study, only emotional forms of harsh parenting.

Reply: Now the sentence of “However, this study found that any forms of parental maltreatment and parental harshness occasioned long-lasting detriments to positive development of immigrant youths across different ethnic origins”  has changed to “However, this study found that parental hostility, emotional rejection, and harsh family interactions with parents experienced by immigrant youths in early adolescence occasioned long-lasting detriments to positive educational development of immigrant youths in terms of development of academic aspiration in late adolescence and educational success in young adulthood across different ethnic origins.” I hope this change can make the presentation of the findings more accurate and precise.

16) The statement in lines 449-452 is not completely correct: emotional neglect is the most prevalent form of child maltreatment, it would be good if the authors would refer to recent large prevalence studies for this statement.

Reply: Now the whole manuscript has been thoroughly revised to let readers know we focus on investigating the effects of parental hostility, emotional rejection, and harsh family interactions with parents, as well as academic aspiration of immigrant youths in late adolescence on successful college graduation of immigrant youth in young adulthood, in which we have stated that this study aims to examine these minor forms of harsh family socialization experiences on adverse educational development of immigrant youths.

17) Hostility, rejection and harsh family interactions are presented as three different variables, which is not consistent with how it was tested in this study (see also my comments in the method section: I do not think it is justified to speak of 'harsh family interactions' as aggregate variable).

Reply: Parental hostility and emotional rejection are investigated as two different types of harsh family socialization experiences of immigrant youths, in which they can be independently or jointly exist in the life of immigrant youths. Hence in this study we want to examine their combined effect as the aggregate of harsh family interactions with parents on educational development of immigrant youths, in which according to the double-whammy thesis we expect harsh family interactions with parents would have the strongest effects on educational development of immigrant youths, which is apparently supported in the current study. In fact, the whole manuscript has now been thoroughly rewritten to make the respective investigations of parental hostility, emotional rejection, and harsh family interactions with parents in contribution to educational development of immigrant youths more justifiable.

18) The limitations of this study are discussed too briefly, and the authors should be more reflective on several important limitations which are currently not mentioned: the operationalization of the 'harsh family interactions' variables and the fact that data collection took place over 20 years ago.

Reply: Now the limitations of the study has been strengthened, which are written as

“Nevertheless, several limitations exist in this study. First, immigrant youths were recruited mainly from the two immigrant-receiving regions of the United States although CILS contains a large and representative sample. Second, this study only examined the effects of parental hostility, emotional rejection, and harsh family interactions with parents in early adolescence of immigrant youths on successful college graduation of immigrant youths in young adulthood through the mediation of their development of academic aspiration in late adolescence. However, other serious types of parental maltreatment, e.g. sexual and physical abuse, and some important intrapersonal constructs of immigrant youths, such as self-esteem and life meaning that may mediate the study relationships, have not examined. Thereby, future research should investigate accumulative effects of different maltreatment experiences of immigrant youths in relation to their educational achievement through the development of multiple cognitive and intrapersonal mediators among immigrant youths. Third, the data of CILS were collected twenty years ago, since then the societal and economic situations of western immigrant-receiving countries have changed great differently [57, 58]. Therefore, more updated data of immigration research are needed to further confirm the relationships between harsh family socialization experiences of immigrant youths and their educational development nowadays. Moreover, as limited by the data structure of CILS, the current study only employed single or few items to measure parental hostility, emotional rejections, and harsh family interactions with parents, which is a major limitation of the current study although some existing research supports the validity of using this measurement approach to tap on minor forms of parental maltreatment [18, 27]. Lastly but not least, as the data of CILS did not include non-Hispanic White youths who distinctively differ to immigrant youths culturally, economically, and socially, cross-population validity and comparison are hence impossible in this study. In all, it is believed that if timely and adequate interventions and service supports are provided to harshly disciplined and maltreated immigrant youths and their families, these youths can also be thriving academically and socially.” (Lines 1703-1779 in the part of “Conclusion”)

19) The authors did a good job explaining the relevance of their study and translating the study's results into implications for society and future studies.

Reply: Thank you again.

As our responses to Reviewer 1 are also relevant to what Reviewer 2 concerns, hence we also enclosed our responses to Reviewer 1 here:

For Reviewer 1

1) The manuscript should be adequately rationalized within the text of the manuscript connecting previous classical and recent research with the focus of the study, specially about parenting.

Reply: Now, the theoretical framework of the manuscript has been thoroughly rewritten, which include “connecting previous classical and recent research with the focus of the study, specially about parenting.” For this, please refer the parts of ““Harsh family interactions with parents and educational achievement” and “The mediation of academic aspiration of immigrant youths”.

2) An important question that should be considered are the measures of parental hostility and emotional rejection. Authors should add more details about the items of parental hostility. It is difficult to find the specific measure, so it should be included the items. The measures seem to be like those about parental responsiveness (lower parental responsiveness). It seems that parental hostility does not include sexual or physical abuse, but the items should be included specifically.

Reply: Now parental hostility and emotional rejection as well as their aggregate as harsh family interaction with parents in relation to academic aspiration and college graduation of immigrant youths in young adulthood have been explained, constructed, and justified in details. For details, please refer to the parts of ““Harsh family interactions with parents and educational achievement” and “The mediation of academic aspiration of immigrant youths”. However, the current study aims to investigate how the minor forms of parental harsh discipline and maltreatment, such as parental hostility and emotional rejection as well as their aggregate as harsh family interaction with parents, which have been received less research attention compared to investigations in parental physical and sexual abuse. In fact, as the dataset of CILS was not planned to investigate family violence and child abuse but the life transformation of immigrant youths after immigration to the United States, hence it does not allow to include those serious child maltreatment constructs, which now has elaborated in the manuscript as its limitations.

3) Within the text of the manuscript, authors should characterize the parental practices of parental hostility and emotional rejection within the model of family socialization based on two dimensions (i.e., responsiveness and demandingness). The dimension of responsiveness represents parental love, approval, acceptance, and support (Gimenez-Serrano et al., 2022). Parental practices of rejection and hostility are characterized by lower levels of responsiveness (Fuentes et al., 2015; Garcia et al., 2020). Overall, results revealed the negative impact of parenting characterized by poor responsiveness (Garcia & Gracia, 2009; Yeung, 2021).

Reply: Now the framework of family socialization model has been put in the study of parental hostility, emotional rejection, and harsh family interactions with parents to explain why and how parental hostility, emotional rejection, and harsh family interactions with parents would negatively affect educational success and academic aspiration of immigrant youths, which are written:

“Correspondently, family transmission model evinces that parents are the main socialization figures responsible to provide care, support, resources, and social and cultural capitals necessary for cognitive and healthy development of their offspring [3]. However, harshly disciplined and maltreated youths even experienced by the minor forms of parental hostility, emotional rejection, or harsh family interactions with parents in family may signify their insufficiency in parental educational involvement and support as well as acquisition of necessary learning resources and guidance to help them overcome academic challenges for achieving educational success [11, 12], as a result posing them poorer educational development. This is especially important for immigrant youths as they are generally living under the conditions of family poverty and weak social supports and resources [31, 32]. Yet, although limited pertinent research has reported the harmful impacts of parental hostility and rejection on behavioral and psychological adjustments of children and youths, little is known about how these negative family socialization experiences may negatively affect educational development of immigrant youths. This is research-worthy from the model of family socialization that youths experiencing harsh discipline and maltreatment implies their receiving low parental responsiveness and support and high parental negativity [22, 33, 34], which are harmful to youth health development educationally and socially. In their meta-analysis, Khaleque [26] found that paternal hostility/aggression significantly associated with psychological maladjustment and negative personality dispositions of children across ethnicities, cultures, and geographical boundaries. Similarly, in their latest study, Backman, Laajasalo, Jokela and Aronen [23] found that parental hostility and lower maternal warmth were significantly predictive of psychopathic behaviors of offending adolescents. More recently, Lee and Mun [35] reported that parental rejection was positively related to cyberbullying behaviors of Korean children and youths of perpetration in a representative sample from the 2019 Korean Children and Youth Panel Survey (KCYPS), in which the study relationship was independent and sequentially mediated by children’s depression and smartphone addiction. Academically, Putnick, Bornstein, Lansford, Malone, Pastorelli, Skinner, Sorbring, Tapanya, Tirado, Zelli, Alampay, Al-Hassan, Bacchini, Bombi, Chang, Deater-Deckard, Di Giunta, Dodge and Oburu [8] found that parental rejection was longitudinally predictive of lower school scores among primary school children across nine countries. Moreover, Ryan, Jacob, Gross, Perron, Moore and Ferguson [20] investigated 732,828 youths born between 2000 and 2006 in Michigan public schools and found that those of child protective service involvement (CPS) exhibited significantly lower math and reading scores, grade retention, and receipt of special education than their peers without CPS involvement. More relevantly, Welsh, et al. [36] found that college students who experienced a history of maltreatment significantly exhibited lower first-semester GPA and poorer college adaptation.” .”(Lines 110-201 in the part of “Harsh family interactions with parents and educational achievement”)

“As academic aspiration of youths has been found to be shaped by the process of family socialization and experiences in early years [14, 15], it is believed in this study that the academic aspiration of immigrant youths in late adolescence not only will directly affect their educational achievement in young adulthood [12, 41, 42], but also may mediate the effects of parental hostility, emotional rejection, and harsh family interactions with parents on their later educational success in young adulthood [17, 38]. In a longitudinal study by Hentges and Wang [14], they found that harsh parenting in 7th grade was significantly related to students’ lower GPA in 11th grade through the mediation of compromised academic values in 8th grade. Moreover, Seginer and Mahajna [17] reported that perceived positive parenting significantly and positively predicated Muslim youths’ academic aspiration for higher education in 11th-grade that in turn mediated the relationship between parenting and academic performance. In her qualitative study, Morton [15] explored how maltreated experiences of foster youths might incur cognitive and behavioral difficulties in relation to their compromised educational achievement and she found that maltreated youths generally held lower academic aspiration that would seriously harm their success of college education. Therefore, in this study, we expect that immigrant youths’ experiences of parental hostility, emotional rejection, and harsh family interactions with parents in early adolescence may negatively impair their development of academic aspiration in late adolescence and successful college graduation in young adulthood, in which the development of immigrant youths’ academic aspiration in late adolescence would mediate the relationships of immigrant youths’ experiences of parental hostility, emotional rejection, and harsh family interactions with parents in early adolescence and their successful college graduation in young adulthood.”(Lines 529-551 in the part of “The mediation of academic aspiration of immigrant youths”)

4) Additionally, the difficulties of immigrant students should be considered more detailed, being immigrant might be related to some problems such as less academic engagement compared to native students (Veiga et al. 2021). The same is true for children in risk neighborhood who had more problems than those from middle-class (Sandoval et al., 2022).

Reply: Right, the current study has mentioned the disadvantaged family and contextual situations that negatively affect educational success of immigrant youths, which include:

“Conspicuously, immigrant youths compared to their non-immigrant counterparts are at greater risk of being harshly disciplined and maltreated by their parents due to lower parental education, disadvantaged social status, and cultural differences [11]. This is correspondent with what Romano, et al. [12] mentioned “socioeconomic disadvantage is important because such stressful conditions as poverty often compromise effective parenting practices (p. 433).” However, no longitudinal research today has been conducted to investigate the negative effects of parental hostility, emotional rejection, and their aggregate as general harsh family interactions with parents experienced by immigrant youths on their educational success in young adulthood. This is important as obtaining a college degree connotes immigrant youths a life chance of upward social mobility, which resonates with Portes, et al. [13] mentioned “(w)ithout the costly and time-consuming achievement of a university degree, such dreams are likely to remain beyond reach (p.1081).””(Lines 62-74 in the part of “Introduction”)

“Correspondently, family transmission model evinces that parents are the main socialization figures responsible to provide care, support, resources, and social and cultural capitals necessary for cognitive and healthy development of their offspring [3]. However, harshly disciplined and maltreated youths even experienced by the minor forms of parental hostility, emotional rejection, or harsh family interactions with parents in family may signify their insufficiency in parental educational involvement and support as well as acquisition of necessary learning resources and guidance to help them overcome academic challenges for achieving educational success [11, 12], as a result posing them poorer educational development. This is especially important for immigrant youths as they are generally living under the conditions of family poverty and weak social supports and resources [31, 32].” (Lines 110-175 in the part of “Harsh family interactions with parents and educational achievement”)

“Accordingly, as immigrant parents are generally of disadvantaged social status, economic deprivation, and cultural differences, they may be more prone of employing harsh discipline and unsupportive parenting [11, 27, 37]. Therefore, it is expected that experiencing parental hostility, emotional rejection, and/or harsh family interactions with parents in family is more common for immigrant youths, which is believed to detriment their educational achievement in adulthood, such as successful college graduation with a four-year undergraduate degree . As such, due to the more disadvantaged and difficult family socialization environment of immigrant youths encountered and the importance of educational success for their social mobility in adulthood as compared to their better-off local counterparts [38, 39], it is theoretically and empirically important to study how immigrant youths’ experiences of parental hostility, emotional rejection, and harsh family interactions with parents in early adolescence may adversely affect their later successful college graduation in young adulthood.”( Lines 203-215 in the part of “Harsh family interactions with parents and educational achievement”)

5) Parental socialization ends when the child arrives to the adult age. The results confirm other previous studies about the long-term impact of parenting with adult children (Candel et al., 2020). Additionally, the statistical power should be considered more detailed as strong point of the study (Faul et al., 2007; Perez et al., 1999).

Reply: Yes, and agree. For power test, it is mainly aimed to ensure a sample size that is large enough for the probability of rejecting the null hypothesis when, in fact, it is false and avoiding a Type II error in order to truly confirm the alternative hypothesis when the null hypothesis is false. In fact, when the sample size increases the power ensues to increase, especially for the samples drawn from representative sampling procedures, like the representative sample used in the current study (N=3344). Even using Z-test of power analysis by setting alpha level at 0.01 and 1-beta error probability level at 0.99, the strictest standard of power test for sample size, the result shows that N=2195 is well enough to ensure the power of rejecting the null hypothesis and confirming the alternative hypothesis. Hence, as it is a well-known fact regarding the assurance of power representative samples, I think it is less relevant to especially highlight the relationship between power and sample size in the current study.